# Variability of nitrogen oxide emission fluxes and lifetimes estimated from Sentinel-5P TROPOMI observations

Kezia Lange[1], Andreas Richter[1], and John P. Burrows[1]

[1]Institute of Environmental Physics, University of Bremen, Bremen, Germany

**Correspondence:** Kezia Lange (klange@iup.physik.uni-bremen.de)

**Abstract.** Satellite observations of the high-resolution TROPOspheric Monitoring Instrument (TROPOMI) on Sentinel-5 Precursor can be used to observe nitrogen dioxide ($NO_2$) at city scales, to quantify short time variability of nitrogen oxides ($NO_x$) emissions and lifetimes on a daily and seasonal basis. In this study, two years of TROPOMI tropospheric $NO_2$ columns, having a spatial resolution of up to $3.5\,km$ x $5.5\,km$, have been analyzed together with wind and ozone data. $NO_x$ lifetimes and emission fluxes are estimated for 50 different $NO_x$ sources comprising cities, isolated power plants, industrial regions, oil fields and regions with a mix of sources, distributed around the world. The retrieved $NO_x$ emissions are in agreement with other TROPOMI based estimates, reproduce the variability seen in power plant stack measurements, but are in general lower than the analyzed stack measurements and emission inventory results. Separation into seasons shows a clear seasonal dependence of $NO_x$ emissions with in general the highest emissions during winter, except for isolated power plants and especially sources in hot desert climates, where the opposite is found. The $NO_x$ lifetime shows a systematic latitudinal dependence with an increase in lifetime from two to eight hours with latitude but only a weak seasonal dependence. For most of the 50 sources including the city of Wuhan in China, a clear weekly pattern of $NO_x$ emissions is found with weekend-to-week day ratios of up to 0.5, but with a high variability for the different locations. During the Covid-19 lockdown period in 2020 strong reductions in the $NO_x$ emissions were observed for New Delhi, Buenos Aires and Madrid.

## 1 Introduction

Nitrogen oxides ($NO_x = NO + NO_2$) play a key role in atmospheric chemistry, air quality and climate. They are emitted into the atmosphere by both natural processes and human activity. Natural sources include lightning, microbial processes in soils and naturally occurring wildfires. The dominant source of $NO_x$ is fossil-fuel combustion, from traffic, residential heating, cooking and the industry and energy sectors. These sources are concentrated in cities and urban areas. In addition, biomass burning and the use of fertilizers are also significant sources of $NO_x$ (Jacob, 1999; Seinfeld and Pandis, 2006; Stocker, 2014).

These sources predominantly release nitrogen monoxide (NO). Nitrogen dioxide ($NO_2$) is released in smaller amounts but is rapidly produced in the atmosphere where NO reacts with ozone ($O_3$). During the day $NO_2$ is photolyzed, reforming NO and producing an oxygen atom O, which forms $O_3$ in a termolecular reaction (Seinfeld and Pandis, 2006).

$$NO + O_3 \rightarrow NO_2 + O_2 \tag{R1}$$

$$NO_2 + h\nu \rightarrow NO + O \tag{R2}$$

$$O + O_2 + M \rightarrow O_3 + M \tag{R3}$$

In the troposphere $NO_x$ are precursors of the health hazard and greenhouse gas $O_3$ but $NO_2$ is also an important health hazard
in its own right (Jacob (1999), and references therein). Consequently, monitoring and understanding its behavior is of particular
importance in cities and urban agglomerations, where high $NO_x$ emissions from multiple sources are found in combination
with high population density. As a result, a large part of the population is exposed to polluted air (Molina and Molina, 2004).
$NO_x$ are short-lived in the atmosphere with a lifetime of several hours in the boundary layer during daytime. This explains in
part the high spatial and temporal variability observed for $NO_2$. Other factors leading to large concentration gradients in $NO_2$
are the heterogeneous distribution of sources and variations of meteorological parameters, such as wind speed, temperature and
illumination, which play key roles in determining the atmospheric lifetime and dilution of $NO_2$ (Beirle et al., 2003; Stavrakou
et al., 2008).

Measurements of $NO_2$ with adequate spatial and temporal resolution are required to assess and compare the variability of
$NO_x$ emissions and lifetimes around the world. Provided that tropospheric $NO_2$ columns retrieved from satellite sensors have
sufficient spatial resolution, the typically short daytime lifetime of $NO_x$ in urban agglomerations provides an opportunity
to disentangle and quantify local sources of $NO_x$ and their variations over time. The TROPOspheric Monitoring Instrument
(TROPOMI) on Sentinel-5 Precursor (S5P), which was launched in October 2017, provides higher spatial resolution than its
predecessors as for example the Ozone Monitoring Instrument (OMI). Tropospheric columns of $NO_2$ retrieved from TROPOMI
thus offer the best opportunity so far to deconvolve urban daytime sources of $NO_x$ (Veefkind et al., 2012; Griffin et al., 2019).
Making use of the $NO_2$ tropospheric columns from TROPOMI, but also from its predecessors, a number of studies on the
variability of $NO_x$ have been carried out.

The high spatial resolution of TROPOMI makes it possible to identify much better $NO_x$ point sources and quantify their emis-
sions. This provides the opportunity to control emission inventories and improve on them (Beirle et al., 2019, 2021).

Investigation of seasonal emission estimates enables the disentanglement of the $NO_x$ sources in a region and the identification
of their individual contributions (van der A et al., 2008). The analysis of TROPOMI data for Paris showed that residential
heating, rather than transport emissions, is the main source of $NO_x$ in winter. This is not well accounted for in current emission
inventories, in which residential heating is underestimated during winter and overestimated during summer (Lorente et al.,
2019).

$NO_x$ emissions also change from day to day, as a result of the human behavior, especially between work and rest days. Such
patterns are readily identified (Beirle et al., 2003). A recent study by Stavrakou et al. (2020) investigated the weekly $NO_2$ cycle
and its trends using the long-term satellite observations of OMI and one year of TROPOMI $NO_2$ column measurements. Sig-
nificant weekly cycles were found. A weakening trend over Europe and the US could be observed. The opposite behavior was

found for regions with increasing $NO_x$ emissions. The decreases or respectively increases in the contribution of anthropogenic $NO_x$ emissions to the observed $NO_2$ levels are thus revealed. With TROPOMI data it is possible to use relatively short periods of data to analyze day to day variability. For Chicago a clear weekend effect with reduced $NO_x$ emissions of 30 % during weekends was found analyzing one season of TROPOMI measurements (Goldberg et al., 2019). Lorente et al. (2019) investigated the day-to-day variability of $NO_x$ emissions in Paris for individual days, with the highest emissions on cold weekdays and the lowest on warm weekend days.

Beginning in early 2020, first China and subsequently the majority of countries around the world took containment measures to limit the spread of Covid-19. These measures resulted in significant changes of human behavior with reductions in traffic density and industrial activity and consequently anthropogenic emissions. Temporally fine resolved emission estimates help to identify the different potential origins of such changes. This provides an approach and an opportunity to better quantify source contributions and to distinguish between different anthropogenic and natural sources of $NO_x$. Several recent studies have analyzed satellite measurements of $NO_2$ columns, and report substantial decreases in the $NO_2$ tropospheric column in early 2020 over China (Liu et al., 2020; Bauwens et al., 2020), northern Italy, South Korea and the United States (Bauwens et al., 2020). However, because of the high variability of $NO_2$ columns these reductions cannot be simply directly attributed to a decrease in $NO_x$ emissions resulting from the Covid-19 containment measures. The tropospheric column of $NO_2$ is influenced by behavioral patterns of anthropogenic activity, seasonality, and meteorology. To account for meteorology, which led to lower $NO_2$ values in spring 2020, as compared to spring 2019 in North America, Goldberg et al. (2020) combined TROPOMI $NO_2$ column data with meteorological data and a chemical transport model and calculated normalized $NO_2$ changes that provide a better representation of Covid-19-related $NO_x$ emission changes.

In addition to $NO_x$ emissions, $NO_x$ lifetimes are determined from tropospheric $NO_2$ columns (Leue et al., 2001; Beirle et al., 2003; Kunhikrishnan et al., 2004). The $NO_x$ lifetime in daylight depends on the rate of loss of $NO_x$ which is attributed primarily to the reaction of the hydroxyl radical (OH) with $NO_2$ to form $HNO_3$. Consequently, there is a nonlinear relationship between the tropospheric concentrations of OH and $NO_x$, which again depends in a complex fashion on the $NO_x$ concentration (Valin et al., 2013; Stavrakou et al., 2008). Laughner and Cohen (2019) showed that $NO_x$ lifetime has changed significantly between 2005 and 2014 in North American cities, changes being of the same order of magnitude as those in $NO_x$ emissions. Another important factor influencing $NO_x$ lifetime is the actinic radiation photolyzing $NO_2$, which varies diurnally and is modulated by the presence of clouds. Shorter lifetimes at smaller solar zenith angles in summer or at lower latitudes due to higher photolysis frequencies are expected (Stavrakou et al., 2020). Typical lifetimes of $NO_x$ range from two to eight hours in polluted air masses and extend to about a day for cleaner more rural background concentrations for which nighttime chemistry also has to be considered (Beirle et al., 2011; Valin et al., 2014; de Foy et al., 2014; Seinfeld and Pandis, 2006). Studies for winter months and especially for winter months at higher latitudes are limited, analyses with the chemical transport model GEOS-Chem show a tendency towards longer lifetimes of about one day (Martin et al., 2003; Shah et al., 2020).

In this study, we use the method first developed by Beirle et al. (2011) and refined by later studies (Pommier et al., 2013; Valin et al., 2013) to estimate the $NO_x$ emissions and lifetimes from TROPOMI $NO_2$ column amounts. Studies of $NO_x$ emissions and lifetimes have been reported using this method for OMI data, which has coarser spatial resolution, for a limited number of

sources and using long periods of data (Ialongo et al., 2014; de Foy et al., 2015; Lu et al., 2015; Liu et al., 2016). Other studies, which applied this method to TROPOMI data are reported by Goldberg et al. (2019) or Lorente et al. (2019). In this work, we provide the first study of $NO_x$ emissions and lifetimes for a large data set, comprising 50 $NO_x$ source regions. These are well distributed over the world, consisting of cities, isolated power plants, industrial regions, oil fields and regions with a mix of sources. We show that the spatial resolution and good signal-to-noise ratio of TROPOMI $NO_2$ columns allow to investigate two years of data and additionally divide and study the following periods: Seasons, weekend and working days, and times before and during Covid-19 restrictions to examine short-term variability. The focus of our investigation is to assess the variability of $NO_x$ emissions and lifetimes in space and time during the period of observation.

## 2 Data

### 2.1 TROPOMI $NO_2$ tropospheric vertical column

In October 2017, the Copernicus Sentinel-5 Precursor (S5P) satellite with the TROPospheric Monitoring Instrument (TROPOMI) on board was launched into a sun-synchronous orbit at 824 km altitude. TROPOMI comprises a hyperspectral spectrometer measuring radiation in the ultraviolet (270 nm - 320 nm), visible (310 nm - 500 nm) and infrared (675 nm - 775 nm, 2305 nm - 2385 nm) spectral regions (Veefkind et al., 2012). The two-dimensional CCD detectors measure spectra in 450 separate viewing directions across the 2600 km swath with an integration time of approximately one second. This results in a high spatial resolution of 3.5 km x 7 km in nadir with little variation across the swath. In August 2019, the pixel size was reduced further to 3.5 km x 5.5 km by reducing along track averaging. One orbit around the Earth takes about 100 minutes, which, in combination with the wide swath, results in a daily global coverage. At higher latitudes, on some days up to three overpasses, with approximately 100 minutes in between the measurements, are available. In the rare case of three overpasses, two of them are on the outer edge of the swath. The equator overpass time of S5P is 13:30 mean local solar time in the ascending node.

In this study, the operational $NO_2$ product of TROPOMI is used (van Geffen et al., 2018). The $NO_2$ product from TROPOMI extends $NO_2$ measurements from earlier instruments such as GOME (1995-2011, (Burrows et al., 1999)), SCIAMACHY (2002–2012, (Bovensmann et al., 1999)), GOME-2 (since 2007, (Munro et al., 2006)), OMI (since 2004, (Levelt et al., 2006)) and OMPS (since 2011, (Dittman et al., 2002)). These instruments had increasing spatial coverage and resolution, OMI being the first instrument with daily global coverage. However, with a resolution of 13 km x 24 km at nadir, its spatial resolution is one order of magnitude poorer than that of TROPOMI in the center of the swath and much lower at the edges. Differences in resolution are larger at the edges, which is due to the fact that for TROPOMI always averages of two pixels in the center of the scan, but not at the edges, are used to compensate for the geometric effect caused by the slanted view to the ground.

The level-1b spectra measured by TROPOMI are analyzed with the Differential Optical Absorption Spectroscopy (DOAS) technique in the fitting window 405 nm - 465 nm. The retrieved $NO_2$ slant column densities are separated into a stratospheric and a tropospheric part based on data assimilation by the TM5-MP global chemistry transport model. The resulting tropospheric slant columns are then converted to tropospheric vertical columns by tropospheric air mass factors (AMFs) using a

look-up table of altitude-dependent AMFs, $NO_2$ vertical profiles from the TM5-MP model, the OMI climatological surface albedo and cloud characteristics derived using the FRESCO algorithm (van Geffen et al., 2018). The final product is the tropospheric vertical column, defined as the vertically integrated number of molecules per unit area between the surface and the tropopause.

In this study, we use the operational level-2 TROPOMI tropospheric vertical column $NO_2$ product from March 2018 to November 2020, the first two years of data for the general study, and additional months from 2020 for the Covid-19 impact study. This includes the reprocessed (RPRO) and offline mode (OFFL) data of version V01.00.01 to version V01.03.02. Changes between the versions before V01.04. are minor and the data products are comparable and can be mixed. V01.04. was activated on 02 December 2020 and implemented major changes, which led to a substantial increase in the tropospheric $NO_2$ column

over polluted areas for scenes with small cloud fractions. We only use the data up to end of November 2020 for our analysis to ensure better comparability, since mixing data from before version V1.04. and after is not recommended. Further major updates in the TROPOMI $NO_2$ retrieval algorithm have been made in V02.02. released on 1 July 2021 (van Geffen et al., 2021). A complete mission reprocessing will be performed to deliver a harmonized data set (Eskes and Eichmann, 2021). Data from 30 April 2018 onwards is freely available on https://s5phub.copernicus.eu/. Data before the 30 April is only available on the

S5P Copernicus Expert Hub and therefore not publicly accessible. The data with a spatial resolution of up to $3.5\,\mathrm{km}$ x $5.5\,\mathrm{km}$ is oversampled to a finer resolution of $0.01°$ x $0.01°$. Each TROPOMI ground pixel is accompanied by a quality assurance value (qa_value). The qa_value ranges from zero (error, no output) to one (no errors or warnings) and indicates the quality of the processing and retrieval result. Based on the recommendation by van Geffen et al. (2018), measurements with a qa_value lower than 0.75 are filtered and not used. A qa_value of 0.75 removes problematic retrievals, errors, partially snow/ice covered

scenes and measurements with cloud radiance fractions of more than $50\,\%$.

## 2.2 Wind data

In addition to the TROPOMI $NO_2$ data, wind speed and direction are required to determine the $NO_x$ emissions and lifetimes. The wind data used is provided by the European Centre for Medium-Range Weather Forecast (ECMWF), ERA5 reanalysis

hourly data with a horizontal resolution of $0.25°$ x $0.25°$ on model levels. To merge the wind data in space and time with the TROPOMI observations, they are interpolated to the overpass time and oversampled to the same $0.01°$ x $0.01°$ resolution as the TROPOMI data. The chosen height of wind data can be critical for wind speed and direction, because of increasing wind speeds and turning with height, and therefore also for $NO_x$ emissions and lifetime estimates. Beirle et al. (2011) investigated the dependence of the estimated $NO_x$ emissions and lifetimes on the wind data level height by comparing the results calculated

with wind fields averaged from ground up to $200\,\mathrm{m}$, $500\,\mathrm{m}$ and $1000\,\mathrm{m}$ and showed little dependence of the results on the wind level height. The resulting emissions and lifetimes change less than $2\,\%$ ($5\,\%$) on average when using $200\,\mathrm{m}$ ($1000\,\mathrm{m}$) instead of $500\,\mathrm{m}$ and less than $15\,\%$ for individual sources. We use wind data averaged over the boundary layer, with the boundary layer height also provided by ERA5. For the early afternoon overpass of TROPOMI, we assume that the boundary layer is well mixed and wind information averaged over this layer are representative for the $NO_x$ emissions and lifetimes investigated in

this study. Using wind data averaged over the boundary layer instead of using a fix height has the advantage that seasonality in wind data due to the seasonal variability of the boundary layer height is accounted for.

## 2.3 Ozone volume mixing ratios

Ozone volume mixing ratios, for converting the $NO_2$ column measurements to $NO_x$ columns to estimate emissions in terms of $NO_x$, were taken from the ECMWF ERA5 reanalysis. The ozone data has the same resolution as the used wind data, hourly data with a horizontal resolution of $0.25°$ x $0.25°$ on model levels. It is averaged over the boundary layer, interpolated to TROPOMI overpass time and oversampled to the same $0.01°$ resolution as the TROPOMI data, in the same manner as for the wind data.

## 2.4 Emission inventory

For comparison with the calculated TROPOMI $NO_x$ emissions, we use the Emissions Database for Global Atmospheric Research (EDGAR) v5.0 of 2015. The bottom-up emission data is available in $0.1°$ x $0.1°$ gridded resolution. For calculation of $NO_x$ emissions in the specific source regions, the gridded emission inventory data have to be integrated over the area for which the top-down method based on TROPOMI data is sensitive. The EDGAR inventory is based on activity data (i.e. population, energy, fossil fuel production, industrial processes, agricultural statistics), mainly from the International Energy Agency (IEA), corresponding emission factors, national and regional information on technology mix data and end-of-pipe measurements (Crippa et al., 2018, 2019). Uncertainties in bottom-up inventories are inferred from the dependence of emission factors on the fuel type, technology and combustion condition, as well as the low-resolution activity data and emission factors. The uncertainty in $NO_x$ emissions from the latest EDGAR inventory version (v4.3.2) varies from $17\%$ to $69\%$ for different regions (Crippa et al., 2018). The limited temporal coverage of bottom-up emissions results in additional uncertainties. 2015 is the most recent year available in EDGAR v5.0, which does not reflect the recent negative and positive trends, which were found from trend analysis of $NO_2$ column satellite data (Georgoulias et al., 2019). Since a large part of the regions analyzed in this study is located in industrialized and highly populated regions, where $NO_2$ has generally decreased according to (Georgoulias et al., 2019), we anticipate that the TROPOMI estimates for the majority of the analyzed regions are lower than the EDGAR estimates for 2015.

## 3 Method

The method to estimate $NO_x$ emissions and lifetimes from satellite column data builds on the heritage of the method introduced by Beirle et al. (2011) and refined by later studies (Pommier et al., 2013; Valin et al., 2013). We use two years of TROPOMI $NO_2$ tropospheric vertical column data from 01 March 2018 to 29 February 2020 for the general analysis, excluding data for which Covid-19 regulations influenced emissions. For the two Chinese cities included in this study, the analyzed period is limited to 22 January 2020 due to early Covid-19 regulations. The Covid-19 impact study in section 4.5 is based on the months from January to November of 2019 and 2020. The three steps of the analysis are shown in Fig. 1 for the Medupi and Matimba

power plants in South Africa as an example. First, the source region is selected. The choice of sources is described in more detail in section 3.1. The next step is a rotation of the satellite measurements around the selected source with the corresponding ERA5 reanalysis wind data to a common wind direction resulting in an upwind-downwind pattern, which is described in more detail in section 3.2. The NO$_2$ column of each pixel is converted into NO$_x$ column and the mean NO$_x$ distribution is calculated. In the last step, we apply a fit with an exponentially modified Gaussian (EMG) function to the averaged NO$_x$ columns to calculate the NO$_x$ emissions and lifetime (see section 3.3).

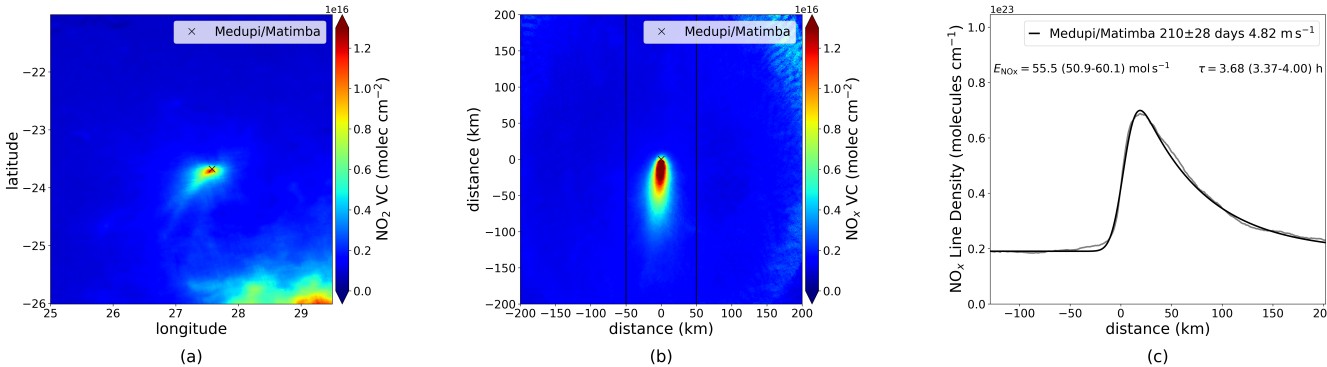

**Figure 1.** (a) Plot of the mean TROPOMI NO$_2$ tropospheric vertical column (VC) from 01 March 2018 to 29 February 2020 in the region of the power plants Medupi and Matimba in South Africa for days with wind speed $> 2\,\mathrm{m\,s}^{-1}$. (b) In this plot each NO$_2$ tropospheric VC is converted to an estimate of NO$_x$ VC and is rotated with its wind direction around the centre of the source (cross) to an upwind-downwind pattern. The black lines indicate the sector of $\pm 50\,\mathrm{km}$ around the source relevant for the emission and lifetime estimates. (c) NO$_x$ VC integrated perpendicular to the wind direction (NO$_x$ line density) as function of distance to the source calculated for the $\pm 50\,\mathrm{km}$ sector (gray) and fit results (black) with estimated NO$_x$ emissions and lifetime. Emission and lifetime uncertainties are one standard deviation retrieved by the fitting procedure.

## 3.1 Selection of sources

To obtain a representative analysis of the variability of NO$_x$ emissions and lifetimes from urban and industrial areas around the globe, we have used the following guideline for the selection of targets. The targeted emission sources need to be well distributed around the world to provide information about the seasonal variations, climatic conditions, latitudinal dependence and weekday behavior of the NO$_x$ lifetimes and emissions. By visually inspecting the global mean NO$_2$ tropospheric vertical column distribution from March 2018 to February 2020 (shown in Fig. 2), we selected 50 target regions, which are marked with red circles, for the calculation of NO$_x$ emissions and lifetimes. The selected sources comprise a mix of cities with predominantly domestic and transport emissions e.g. Paris (France), cities with more industrial emitters e.g. Chelyabinsk (Russia), power plants like Medupi and Matimba (South Africa) or oil refinery regions like the Sarir Field (Libya). The method works best for isolated point sources with a high contrast between source and background. Regions having few clouds are preferred to maximize the number of satellite observations. In addition, a local meteorology for the target region, having rather

homogeneous wind patterns, facilitates the detection of the outflow patterns, which is further enhanced by filtering out data with wind speeds of less than $2\,\mathrm{m\,s^{-1}}$ (Beirle et al., 2011). Some, initially promising sources like Santiago de Chile, were omitted.

This was a result of their location in coastal and mountainous regions with inhomogeneous terrain, resulting in inhomogeneous wind patterns, which are more difficult to interpret and lead to larger uncertainties in the $NO_x$ emission rates and lifetimes. For this study, many of the sources with high $NO_2$ signal in China were not used, because the influence of large $NO_x$ emitting sources nearby resulted in low contrast of the $NO_2$ column amount between the target and the local background. For such conditions, other methods to determine $NO_x$ emission rates and lifetimes are more appropriate (Liu et al., 2016). Despite

efforts to select a broad range of regions in many respects, some selection bias could have been introduced, for example, by not analyzing regions with low wind speeds, mostly cloudy conditions or areas in regions with many emission sources and higher background levels.

If the conditions are favorable (e.g clear-sky, homogeneous and strong wind condition) even a single overpass can deliver

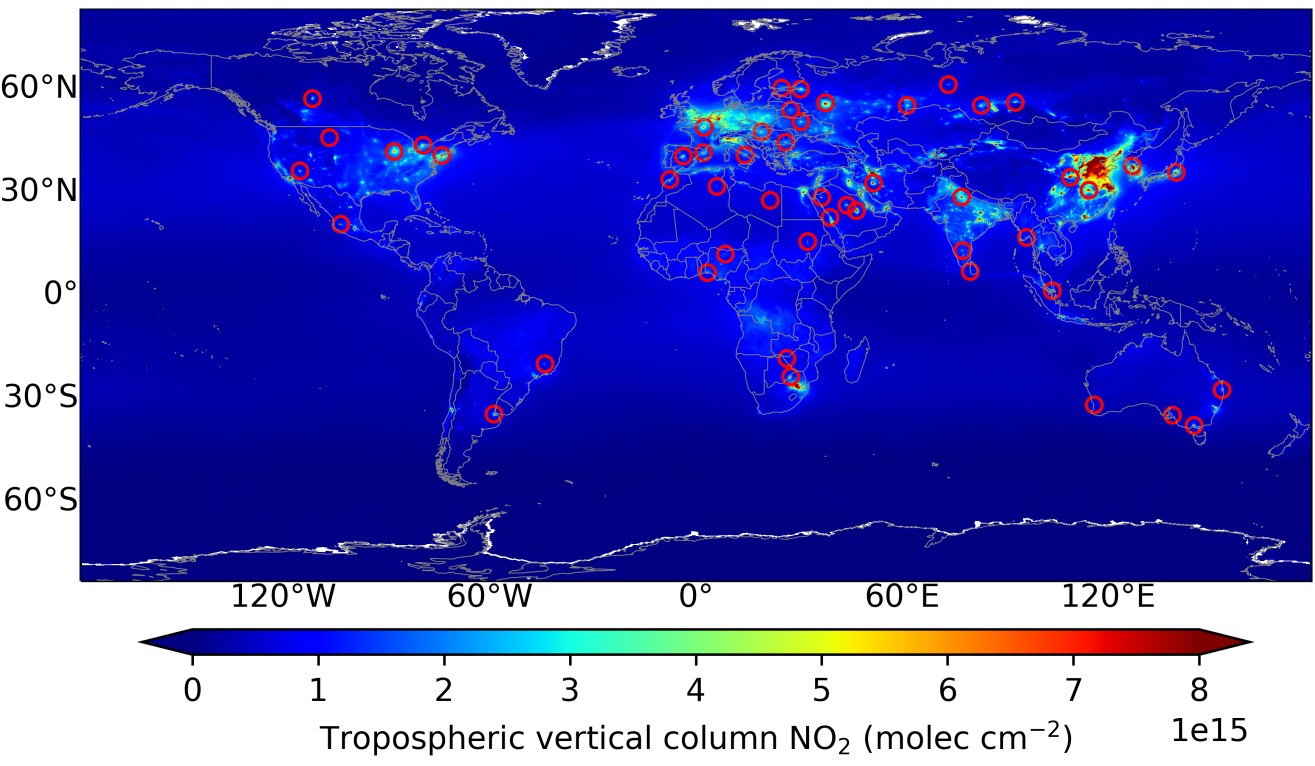

**Figure 2.** $NO_2$ tropospheric vertical column of the level-2 Sentinel-5P TROPOMI product from March 2018 to February 2020. Red circles mark $NO_x$ emission sources analyzed in this study. All analyzed source regions are listed with more details in the appendix table A1.

valuable results for some regions (Lorente et al., 2019; Goldberg et al., 2019). For analyzing the variability due to seasonal or

220 weekday effects in a more general way, it is useful to have some statistics to prevent the introduction of a bias into the seasonal

analysis by weekday variability. Due to the high spatial resolution and the good signal to noise ratio of the TROPOMI data, it is possible to use a data set of only two-years length to analyze successfully regions, which are covered more frequently by clouds or have low NO$_2$ signals. Nevertheless, not every source region is used for every analysis in this study because of data availability after separating data into seasons or week and weekend days. From the 50 targeted regions shown in Fig. 2, eight
are in the southern hemisphere. In order to integrate them into the analysis, they are mirrored in latitude and shifted by six months in season. All source regions are listed in the appendix table A1. Surgut (Russia) is the NO$_x$ source with the highest latitude at 61.25° N, and Singapore (Singapore) is the one closest to the equator at 1.3° N.

One example of the selected source regions is shown in Fig. 1. Figure 1 (a) shows the average of NO$_2$ tropospheric vertical columns for two years of data from 1 March 2018 to 29 February 2020 for days with wind speeds $> 2 \, \mathrm{m \, s^{-1}}$ in the region of
the power plants Medupi and Matimba in South Africa. The NO$_2$ distribution shows an isolated source and its plume which has a high contrast to a low background concentration. In the south eastern part of the map high tropospheric NO$_2$ columns are visible. These originate from the South African Highveld conurbation near Johannesburg.

## 3.2   Rotation technique

To investigate the spatial pattern of NO$_2$ column measurements we combine the approach from Beirle et al. (2011) of a
directional classification to determine the distribution, as a function of downwind distance, with a rotation of each TROPOMI measurement with its merged wind direction around the source to a common wind direction (Pommier et al., 2013; Valin et al., 2013).

Each TROPOMI observation is merged with the ERA5 wind data and rotated with its wind direction about a reference point (centre of source, e.g. the city centre or the power plant site) to a common wind direction, preventing a neutralization of outflow
patterns of opposite wind directions. After the rotation, the TROPOMI measurement points are redistributed along the upwind-downwind direction with the enhancement located in the downwind area of the source reference point and a clear outflow pattern. The measurement points maintain their upwind-downwind character and are analyzed simultaneously, independent of their wind direction. Figure 1 (b) shows the mean of the rotated NO$_x$ tropospheric vertical column for the Medupi/Matimba example, where a pronounced plume with a clear upwind-downwind distribution is found. The black lines indicate a sector of
$\pm 50 \, \mathrm{km}$ around the source. Only data in this sector is used in the next step for the calculation of NO$_x$ emissions and lifetime. As an additional quality filter, only days, where at least 50 % of the ground scenes in this area contain measurements and are not filtered because of clouds or wind speed, are used in the analysis.

## 3.3   Line density calculation

The average outflow pattern of the NO$_2$ tropospheric vertical column with a decay of the signal with distance from the reference
source point reflects transport and nonlinear effects of atmospheric chemistry. Beirle et al. (2011) proposed a model to estimate the NO$_x$ emissions and lifetimes by integrating the mean NO$_2$ columns in the across-wind direction and thereby reducing the two-dimensional maps to one-dimensional so-called line densities with units molecules cm$^{-1}$. In this study, we first converted the NO$_2$ columns for each pixel into NO$_x$ columns to obtain NO$_x$ line densities from which NO$_x$ emissions are calculated

directly (comparable to Beirle et al. (2021)) instead of applying the commonly used fixed [NO$_x$]/[NO$_2$] ratio of 1.32. Assuming that the Leighton photostationary state applies for the polluted air masses investigated, the NO$_2$ is considered a surrogate for NO$_x$ and concentrations of NO and NO$_2$ are coupled by:

$$\frac{[\text{NO}_x]}{[\text{NO}_2]} = 1 + \frac{[\text{NO}]}{[\text{NO}_2]} = 1 + \frac{J_{\text{NO}_2}}{k_{\text{NO}+\text{O}_3} \cdot n_{\text{O}_3}} \tag{1}$$

with $J_{\text{NO}_2}$ the photolysis frequency of NO$_2$ and $k_{\text{NO}+\text{O}_3}$ the rate constant for the reaction of NO with O$_3$ (Seinfeld and Pandis, 2006).

Ozone data is taken from ERA5 and interpolated to the S5P overpass as described in section 2.3.

For clear-sky conditions the photolysis frequency in the boundary layer is parameterized as a function of solar zenith angle (SZA) as proposed by Dickerson et al. (1982):

$$J_{\text{NO}_2} = 0.0167 \exp\left(-\frac{0.575}{\cos(\text{SZA})}\right) (\text{s}^{-1}). \tag{2}$$

The rate constant $k_{\text{NO}+\text{O}_3}$ can in general be well represented by the Arrhenius expression, following the recommendation by Atkinson et al. (2004):

$$k_{\text{NO}+\text{O}_3}(T) = 2.07 \cdot 10^{-12} \exp\left(-\frac{1400}{T}\right) (\text{cm}^{-3}\,\text{molec}^{-1}\,\text{s}^{-1}) \tag{3}$$

with temperature $T$ (in kelvin) taken from the ERA5 reanalysis; hourly data with a horizontal resolution of 0.25° x 0.25°, averaged about the boundary layer, interpolated to TROPOMI overpass time and oversampled to the same spatial resolution.

From the averaged NO$_x$ columns NO$_x$ line densities are calculated by integrating perpendicular to the wind direction. To reduce the influence of possible surrounding sources, the line density is only calculated in a sector around the source (black vertical lines in Fig. 1 (b)), which does not cut the plume and thereby misses emissions. Typical values used for the sector size are between $\pm 15\,\text{km}$ and $\pm 70\,\text{km}$ across the plume depending on its width, and up to $200\,\text{km}$ upwind and up to $400\,\text{km}$ downwind of the source depending on plume length and the presence of other influencing sources. Figure 1 (c) shows the calculated line density as function of distance to the source. The calculated line density is shown in gray and the fit in black. Due to mixing and transport processes, the maximum is shifted in wind direction and the line density curve is steep upwind and less steep downwind with an exponential decay. From this line density curve, NO$_x$ lifetime and emissions can be estimated. The fitting model $M$ as function of distance to the source $x$ is described by (similar to Beirle et al. (2011) supplement):

$$M(x) = E' \cdot (e \otimes G)(x) + B \tag{4}$$

with a convolution of the exponential $e$ and the Gaussian $G$ function scaled by a multiplicative emission factor $E'$ and shifted by a background concentration offset $B$. The exponential function describes transport and chemical decay:

$$e(x) = \exp\left(\frac{-(x - X)}{x_0}\right) \tag{5}$$

with $x > X$ (downwind) and else zero, where $X$ is the location of the apparent source relative to the source reference point and $x_0$ the distance over which the line density decreases by a factor of $e$ ($e$-folding distance). The Gaussian function represents

the broadening of the source by spatial smoothing with the Gaussian function width $\sigma$, which accounts for spatial smoothing caused by the extent of the spatial source, the TROPOMI pixel size and wind variations:

$$G(x) = \frac{1}{\sqrt{2\pi}\sigma} \exp\left(-\frac{x^2}{2\sigma^2}\right). \tag{6}$$

This results in:

$$
\begin{aligned}
M(x) &= E' \cdot \left(\exp\left(\frac{-(x-X)}{x_0}\right) \otimes \frac{1}{\sqrt{2\pi}\cdot\sigma}\cdot\exp\left(-\frac{x^2}{2\cdot\sigma^2}\right)\right) + B \\
&= \frac{E'}{2}\cdot\exp\left(\frac{\sigma^2}{2\cdot x_0^2} - \frac{x-X}{x_0}\right)\cdot\mathrm{erfc}\left(\frac{\sigma^2 - x_0(x-X)}{\sqrt{2}\cdot\sigma\cdot x_0}\right) + B.
\end{aligned}
\tag{7}
$$

The fitted $e$-folding distance $x_0$ and the mean wind speed $w$ from the line density sector is then used to calculate the mean lifetime:

$$\tau = \frac{x_0}{w}. \tag{8}$$

The calculated lifetime includes effects of deposition, chemical conversion and wind advection but must be considered as an effective mean dispersion lifetime. This is because, downwind changes for example due to a changing [NO$_2$]/[NO$_x$] ratio or the effects of non-linearities in the NO$_x$ lifetime in the plume are not considered in the method used.

The multiplicative emission factor $E'$ characterizes the total amount of NO$_x$ near the source and is used together with the mean wind speed $w$ from the line density sector to derive the NO$_x$ flux in $\mathrm{mol\,s^{-1}}$:

$$E_{\mathrm{NO}_x} = \frac{E'}{N_\mathrm{A}}\cdot w \tag{9}$$

with the Avogadro constant $N_\mathrm{A} = 6.02214076\cdot 10^{23}\,\mathrm{mol^{-1}}$.

The described method to calculate NO$_x$ emissions and lifetimes will be named exponentially modified Gaussian (EMG) method. Uncertainties and error bars for NO$_x$ emission and lifetime estimates presented in the following results are based on one standard deviation derived by the EMG fitting procedure and are calculated with error propagation. The total uncertainty of NO$_x$ emissions and lifetimes is influenced additionally by many different error sources, especially the TROPOMI NO$_2$ tropospheric vertical column itself, which are mainly systematic and lead to an overall low bias of the derived NO$_x$ emissions which is discussed in more detail in section 4.6.

## 4  Results and Discussion

The EMG method was applied to the mean TROPOMI NO$_2$ column data of the selected regions and NO$_x$ emissions and lifetimes were calculated. For all available TROPOMI measurements of the two years period from 01 March 2018 to 29 February 2020, and also separated into seasons, working days and weekends and pre Covid-19 times and the Covid-19 pandemic. Not all source regions are used for all analysis due to sometimes poor statistics when separating the two years of data into specific periods.

## 4.1 Comparison to similar studies and emission inventories

The $NO_x$ emission estimates from this study are first compared to results from other recent studies (Beirle et al., 2019; Goldberg et al., 2019; Lorente et al., 2019), which estimated $NO_x$ emissions with TROPOMI data. The studies used for comparison focused only on specific regions, used different periods, and the methods differ slightly but have in common that all used TROPOMI data for their emission calculations. Table 1 compares the $NO_x$ emission estimates for all source regions used in the comparative studies to the emission estimates retrieved in this study. For the study of Goldberg et al. (2019), which uses a time window from May to September 2018, it is possible to estimate and compare the results for the same period, values are given in brackets. Due to different periods and slightly different methods and input parameter, differences are expected but in general the emissions are in reasonable agreement.

Beirle et al. (2019) used modified operational TROPOMI $NO_2$ data, for the period from December 2017 to October 2018

**Table 1.** $NO_x$ emissions estimated for seven source regions are compared to the emissions derived from other studies using TROPOMI data. Values in brackets are derived for the same data period as in Goldberg et al. (2019). Errors are one standard deviation derived by the EMG fitting procedure.

| Source region | $NO_x$ emissions (mol s$^{-1}$) | | | |
| --- | --- | --- | --- | --- |
| | This study | Beirle et al. (2019) | Goldberg et al. (2019) | Lorente et al. (2019) |
| Riyadh | 186.1 ± 7 | 144.8 - 184.5 | — | — |
| Medupi/Matimba | 55.5 ± 4.5 | 37.2 | — | — |
| Chicago | 82.9 (62.3) ± 7 (8) | — | 73 | — |
| New York | 101 (75.5) ± 8 (5) | — | 57.4 | — |
| Toronto | 53.8 (50.8) ± 3.5 (4.5) | — | 51.9 | — |
| Colstrip | 4.6 (4.9) ± 0.2 (0.4) | — | 5.3 | — |
| Paris | 56.2 ± 4.5 | — | — | 53 |

and $NO_x$ emission estimates are based on the continuity equation. For Riyadh $NO_x$ emissions are estimated for the urban area and a greater area of 250 x 250 km$^2$ around Riyadh. Emissions retrieved in this study are slightly higher than those for Riyadh and significantly higher than those for Medupi/Matimba given in Beirle et al. (2019). Possible reasons for this are different wind fields and the conversion from $NO_2$ to $NO_x$. Beirle et al. (2019) used wind information from a fixed vertical level at about 450 m and a fixed $[NO_x]/[NO_2]$ ratio of 1.32 compared to the wind data averaged over the boundary layer and the conversion based on the assumed photo-stationary state with ERA5 ozone data used in this study. For Riyadh the estimated mean $[NO_x]/[NO_2]$ value is 1.41 and therefore slightly higher than the 1.32 used by Beirle et al. (2019). For Medupi/Matimba it is with 1.54 significantly higher and can explain in large part the higher $NO_x$ emissions estimated in this study compared to those in Beirle et al. (2019).

Goldberg et al. (2019) used operational and modified $NO_2$ data from May to September 2018, from which here only the emissions based on the operational data product are used. They found that plumes can be misaligned by up to 30° after rotation

based on the ERA5 wind data. Consequently the authors made a visual inspection of every day and manually adjusted the data. Due to the focus on summer months in Goldberg et al. (2019) and generally lower $NO_x$ emissions in summer than in winter (see also section 4.2), lower values are expected in comparison to the two-year average used in this study. For a better

comparison we added estimates for the same period from May to September 2018 and as expected the agreement improves. Nevertheless, differences remain and are especially significant for New York, possibly the different wind information have a greater influence here.

Lorente et al. (2019) used 36 orbits of the operational $NO_2$ product obtained on 34 different days between February and June 2018 to investigate the $NO_x$ emissions from Paris. Their $NO_x$ emissions of $53\,\mathrm{mol\,s^{-1}}$ agree well with the estimated emissions

for Paris of $56.2\,\mathrm{mol\,s^{-1}}$ from this study.

The comparison indicates that the $NO_x$ emissions from the source regions are in agreement with previous estimates taking into account differences in the analysis techniques.

The retrieved $NO_x$ emissions for the two years of data have also been compared to those reported in the EDGAR emission database. In Fig. 3 the estimated $NO_x$ emissions for the 50 source regions are plotted versus the respective emissions received

from the EDGAR database for the year 2015. Most $NO_x$ source regions have higher emissions in the EDGAR database than estimated by the EMG method in this study. One possible explanation for differences is the different periods the two methods are based on with 2015 for the EDGAR database and March 2018 to February 2020 for this study. In addition, the EMG method is only sensitive to daytime emissions on nearly cloudless days close to the time of measurements, whereas the EDGAR database gives 24-hour annual averages. Another possible explanation is the known low bias of current TROPOMI

tropospheric $NO_2$ columns, as compared to ground based or aircraft measurements of the tropospheric $NO_2$ column. This underestimation is more pronounced for regions with larger $NO_2$ columns (Verhoelst et al., 2021), which is in agreement with our finding that differences to EDGAR are largest for the source regions with high $NO_x$ emissions such as Tokyo, Wuhan, Moscow and New York. This underestimation of TROPOMI will probably be reduced using the reprocessed data set with V02.02 and is discussed in more detail in section 4.6. The estimated $NO_x$ emissions for the Medupi and Matimba power plants

are an exception as they show significantly higher emissions estimated by the EMG method than reported in the EDGAR database. One likely explanation for this observation is that the Medupi power plant was put in operation only in 2015, the year to which the EDGAR database refers to. Consequently, it was not or only partially considered in EDGAR. This also shows the potential of the EMG method to identify $NO_x$ point sources that are not included in emission inventories.

Emissions from power plants in the United States are measured continuously at the stacks and are made available as hourly

mean values by the Environmental Protection Agency's (EPA) Continuous Emission Monitoring System (CEMS). The $NO_x$ emissions for the power plant Colstrip (Montana, USA) are reported in the CEMS database and can also be analyzed with the EMG method based on TROPOMI data. Figure 4 (a) shows a time series of the $NO_x$ emissions reported by the CEMS for the Colstrip power plant from February 2018 to March 2020 (blue) and reported data filtered for days with TROPOMI measurements that were used for the EMG method (green). Assuming that a typical $NO_x$ lifetime is around three hours on

average for the Colstrip power plant, reported $NO_x$ emissions are averaged between 11 to 14 local time, approximately three hours before a typical TROPOMI overpass. The reported $NO_x$ emissions for the Colstrip power plant show a high variability

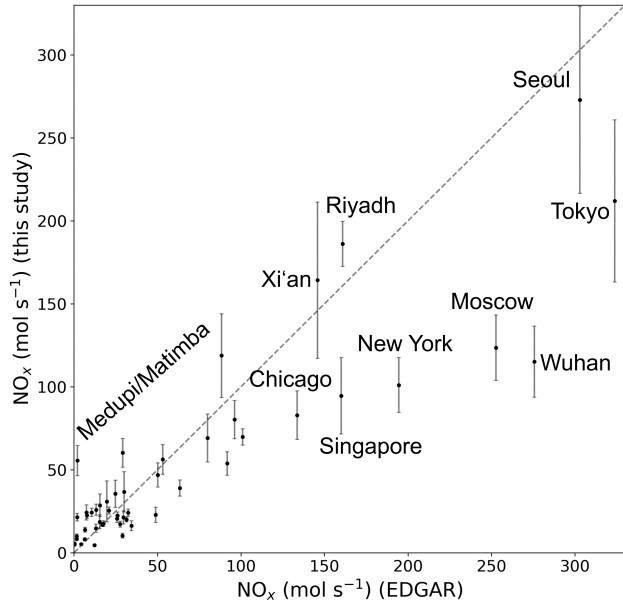

**Figure 3.** $NO_x$ emissions derived from two years of TROPOMI data (01 March 2018 to 29 February 2020) for 50 sources calculated with the EMG method compared to the $NO_x$ emissions derived from the EDGAR emission database (v.5.0, 2015). The dashed line shows the 1:1 ratio. Error bars are the standard deviation derived by the EMG fitting procedure for emission estimates.

over the course of the year.

The EMG method is a widely used method but mostly used for data around summer. In this study we will also extensively use winter data which bring greater uncertainties to the analysis, for example due to longer lifetimes in winter and additionally
data availability is often worse during winter than summer. Therefore, we wanted to verify if the EMG method can reproduce the variability seen in the reported emissions independently of season. For analyzing this variability, we defined seven periods of a range between two to five months during February 2018 to March 2020 (dashed black lines). CEMS data reported on TROPOMI measurement days are averaged for these periods and are shown as horizontal green lines with standard deviation. Estimates from the EMG method for these periods are shown as horizontal red lines with standard deviation derived by the
EMG fitting. The averaged CEMS data and EMG estimates show a similar pattern of $NO_x$ emissions over the year with an underestimation of the CEMS data by the TROPOMI EMG results for all periods. This underestimation could be explained by a loss of $NO_x$ after its measurement in the power plant stack or in the early stage of the $NO_2$ plume formation, and to a large extent by the discussed known low bias of the TROPOMI $NO_2$ data. The low bias is expected to be particularly pronounced for power plants due to wrong AMFs from the TM5 model vertical profiles (Beirle et al. (2019), supplement). Figure 4 (b) shows
a comparison of the two data sets for the seven periods in a scatter plot. The 1:1 line is indicated with the gray dashed line. The solid black line indicates the linear regression, with a slope of $0.4 \pm 0.09$ and a correlation of 0.9, reflecting the underestimation of the CEMS data by the TROPOMI EMG results but also the well matching emission patterns.

From this comparison we can conclude that power plant emissions can vary strongly over time and that the TROPOMI based

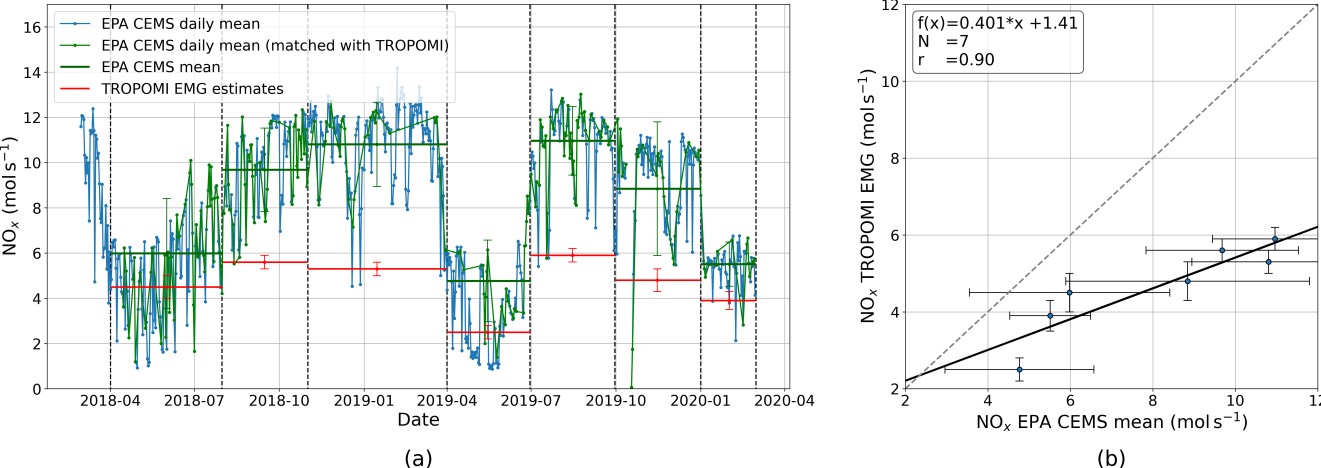

(a)                                                    (b)

**Figure 4.** (a) CEMS $NO_x$ emissions for the Colstrip (Montana, USA) power plant compared to the $NO_x$ emissions estimated by the EMG method, using TROPOMI $NO_2$ data. CEMS data are daily mean values, averaged between 11-14 local time, shown in blue and filtered for days with TROPOMI measurements shown in green. Means over seven selected periods, marked by the dashed black lines, are shown by the horizontal green lines for the CEMS data and red lines for the TROPOMI EMG estimates. (b) Scatter plot of CEMS measured $NO_x$ emissions vs TROPOMI EMG $NO_x$ emission estimates for the seven periods. The 1:1 line is indicated with the gray dashed line. The solid black line indicates the linear regression. Error bars are the standard deviation derived by the EMG fitting procedure for the EMG emission estimates and the monthly standard deviation for the CEMS mean values.

$NO_x$ emissions can be lower by about a factor of two for power plants as already discussed in previous publications (Beirle
et al., 2019). However, from this comparison, we can conclude that the EMG method applied to TROPOMI data at least for this
power plant reproduces the temporal variability seen in the CEMS data reasonably well, and does not show a clear seasonal
bias. Therefore, it can be assumed that the method also gives reasonable results analyzing winter data.

## 4.2   Seasonality of $NO_x$ emissions

To investigate the seasonal variation of $NO_x$ emissions, the two years of TROPOMI data are separated into four seasons
and emissions are calculated separately for each season. Winter months are defined as December to February (DJF, southern
hemisphere JJA), spring as March to May (MAM, southern hemisphere SON), summer as June to August (JJA, southern
hemisphere DJF) and autumn as September to November (SON, southern hemisphere MAM). The two-year data set provides
for each season a maximum of two times three months. Due to cloud cover, inhomogeneous wind patterns and resulting partly
poor statistics after separating into seasons, only 34 of the 50 source regions are analyzed. For Wuhan which is already affected
by COVID-19 regulations when analyzing data from March 2018 to February 2020, the analysis is limited to data from March
2018 to 22 January 2020.

In Fig. 5 the summer-to-winter ratio of the $NO_x$ emissions are plotted for these 34 source regions. For most of the source regions, it was found that the $NO_x$ emissions are higher during winter months than during summer. Naples (Italy), Wuhan (China), Novosibirsk and Krasnoyarsk (Russia) are among the cities with the lowest summer-to-winter ratio of around 0.3 and accordingly clearly higher $NO_x$ emissions in winter than during summer. This is expected in places where domestic heating in winter contributes significantly to $NO_x$ emissions. For some source regions the found ratios are unexpected as for example for Casablanca (Morocco), with a summer-to-winter ratio of $0.32 \pm 0.17$. Due to the location on the Atlantic coast, which mitigates temperature fluctuations, summers are hot but typically not very hot and winters mild, smaller seasonal variation in emissions is expected. No explanation for this observation could be found so far. For some regions, the $NO_x$ emissions are actually higher in summer than in winter, for example Medupi/Matimba in South Africa. Interestingly, the regions with a summer-to-winter ratio larger than one or close to one in Saudi Arabia (Riyadh, Rabigh, Tabuk, Buraidah), Libya (Sarir Field) and Sudan (Khartoum) are all located in hot desert climate regions which are dry and warm all year round, but where especially in summer long heat periods are common. A likely explanation is therefore, that the $NO_x$ emissions are higher in summer than in winter due to higher power consumption caused by the use of air conditioning in summer.

Beirle et al. (2011) reported that the derived $NO_x$ emissions are stable throughout the year for Singapore and Madrid, and that variability found for other cities, which could reflect a change in emissions, was not significant. Most other emission studies focused on summer months only and therefore an investigation of seasonality of $NO_x$ emissions was not possible (Beirle et al. (2019); Goldberg et al. (2019); Ialongo et al. (2014)). Lorente et al. (2019) found the highest $NO_x$ emissions to occur on cold weekdays in February and lowest emissions on warm weekend days in spring 2018. Combined with comparisons to emission inventories the authors concluded that this indicated the importance of the contributions from the residential heating sector to $NO_x$ emissions in winter. Using the full two-year data set, we found a summer-to-winter ratio of $0.51 \pm 0.2$ for Paris, which supports the results of Lorente et al. (2019).

Overall, the investigation of seasonality shows higher $NO_x$ emissions in winter than in summer for the majority of source regions, which is expected for the location of the majority of the selected regions in mid-latitudes and temperate climate. Not all ratios are significantly different from unity, and some results are unexpected, but the trend from the source regions at high latitudes with higher $NO_x$ emissions in winter to the regions in desert climate with higher emissions in summer is plausible. In some part our results differ from those previously reported in the literature (Crippa et al., 2020; Zheng et al., 2018) where generally weaker seasonality is found. This may reflect the use of emission inventories and the analysis of average values for countries in the literature that probably does not reflect the specific situation in cities. The composition of sources for a specific city is expected to be different as that for a country. Further detailed investigations and comparisons are needed to understand the discrepancies. Nevertheless, it should be considered that the uncertainties of the EMG $NO_x$ emission estimates are relatively large and especially the analysis of winter data has not been performed often and is influenced even more by uncertainties (see section 4.1 and 4.6).

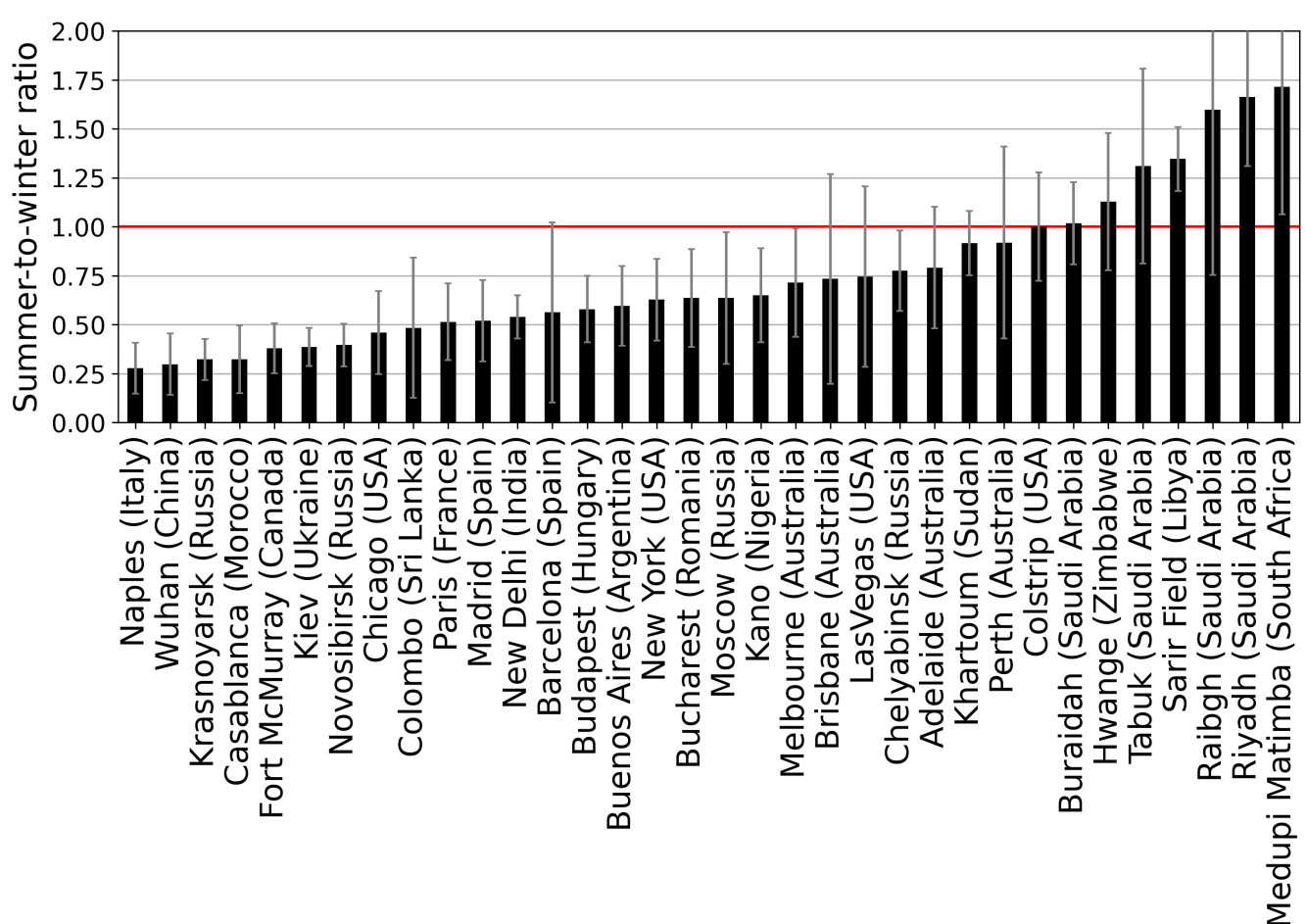

**Figure 5.** Summer-to-winter ratio of retrieved $NO_x$ emission data (01 March 2018 to 29 February 2020). Data is separated into winter (northern hemisphere: DJF, southern hemisphere: JJA) and summer (northern hemisphere: JJA, southern hemisphere: DJF) for 34 source regions. From left to right in increasing ratio. The red line indicates the 1:1 line where summer and winter $NO_x$ emissions are equal, below the line winter emissions are larger and over the line the summer emissions are predominant compared to the winter emissions. Error bars are the standard deviation derived by the EMG fitting procedure for emission estimates.

## 4.3 Latitudinal and seasonal dependence of lifetimes

Another parameter retrieved by the EMG method is the mean effective lifetime of $NO_x$. Thus, in addition to the seasonality of $NO_x$ emissions, the seasonality and latitudinal dependence of lifetimes has been investigated. When data averaged over the two years is considered, lifetimes can be calculated for all 50 source regions. Separating the data set into seasons reduces the number of available data and lifetimes can be estimated for 34 of the in total 50 source regions.

Figure 6 (a) shows the line density in dependence of distance to the source, separated into four seasons for the city Madrid.

The light blue line is the calculated line density and the dark blue one the fitted curve for the winter months (DJF). For the two winters (six months) between 01 March 2018 and 29 February 2020, 25 days could be used for the analysis with a mean wind speed on these days of $5.5\,\mathrm{m\,s^{-1}}$. For winter, this results in $NO_x$ emissions of $84.2 \pm 8.8\,\mathrm{mol\,s^{-1}}$ and a lifetime of $2.41 \pm 0.27\,\mathrm{h}$. The results for spring (MAM) are shown in green, the summer results (JJA) in red, and the autumn months (SON) are shown in yellow. The calculated lifetimes vary between $1.59 \pm 0.16\,\mathrm{h}$ in summer and $2.41 \pm 0.27\,\mathrm{h}$ in winter. The average lifetime over the full two years period is $2.03 \pm 0.15\,\mathrm{h}$.

In Figure 6 (b) estimated $NO_x$ lifetimes for ten source regions are shown as function of season. Only a selection of the sources is shown here, which are a mix of regions from southern and northern latitude, distributed over many latitudes and with different source types. In general these source regions show similar behavior with the longest lifetimes in winter, shortest in summer and quite similar lifetimes in spring and autumn that fall between the summer and winter estimates. Also deviations are visible as could already be seen for the case of Madrid, where the autumn lifetime is much shorter than in spring. Seasonal differences are the strongest for higher latitudes and the lowest for sources the closest to equator. In general, the results are in agreement with values shown in Beirle et al. (2011), where eight different source regions were analyzed yielding lifetimes within a range of two to six hours and a maximum of 8.5 hours during wintertime for Moscow.

The averaged lifetimes over the full two-years period for these 50 source regions as a function of latitude are shown in Fig. 6 (c). As a result of the lower sun and thus reduced photolysis and likely lower OH at higher latitudes, one would expect an increase of $NO_x$ lifetime with latitude (Martin et al., 2003; Stavrakou et al., 2013). This is broadly confirmed by the results, averaged lifetimes increasing from approximately two hours for source regions at low latitudes near the equator to about six to eight hours for source regions at high latitudes of around 60 degrees.

The retrieved lifetimes can be used to estimate atmospheric OH concentrations. Assuming that the decay of $NO_x$ is determined by the termolecular reaction of OH with $NO_2$ with the rate constant $k_{OH+NO_2+M}$ for 298 K and 1 atm reported by Burkholder et al. (2020), the mean OH concentration ranges from $0.3 \cdot 10^7$ - $1 \cdot 10^7\,\mathrm{molec\,cm^{-3}}$ for a range of $NO_x$ lifetimes of two to eight hours. This is in a reasonable range for OH concentrations at midday (Holland et al., 2003; Smith et al., 2006; Lu et al., 2013).

Figure 6 (d) shows in addition to the two years mean also the lifetimes separated by seasons. As result of the reduced number of data points not all source regions (34 out of 50) could be included. All seasons show similar latitudinal dependence as described for the mean lifetime but in addition a seasonal dependence is visible with the shortest lifetimes in summer and longer lifetimes in winter.

We can only detect a weak seasonality of the $NO_x$ lifetime, which is in agreement to Beirle et al. (2011) but significantly lower than would be expected from the analyses with the chemical transport model GEOS-Chem by Martin et al. (2003) and Shah et al. (2020), which showed lifetimes of about one day in winter compared to around six hours in summer. Possible explanations could be not yet sufficient statistics, a clear-sky bias and also the midday observation time of TROPOMI, which all lead to more balanced lifetimes in summer and winter. Furthermore, it should be considered that the total uncertainty for lifetime estimates is larger than just the standard deviation derived by the EMG fit, especially in winter due to longer lifetimes and often reduced data availability (section 4.6). The large data set used in this study reveals a clear but complex latitudinal

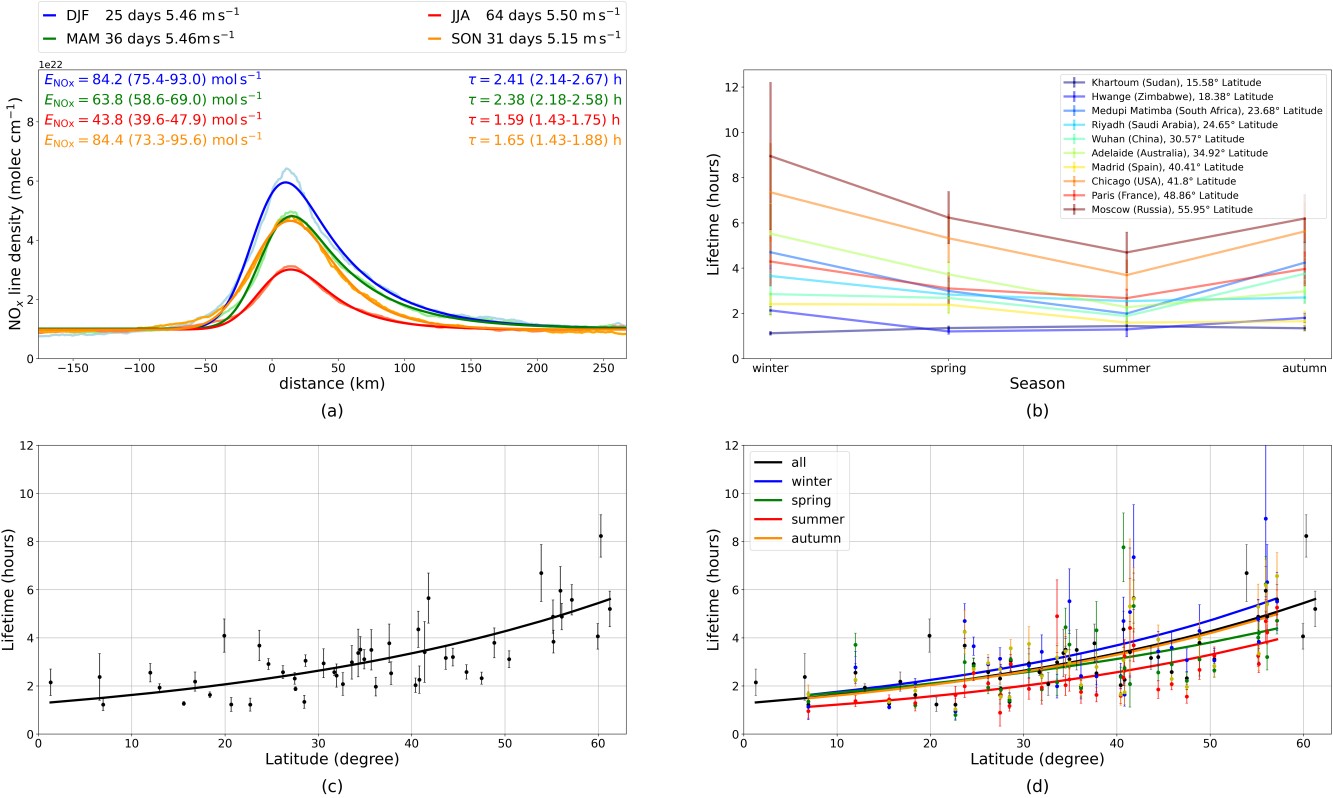

**Figure 6.** (a) The seasonal NO$_x$ line densities for the city of Madrid. The line densities are in light colors and the fit in intense colors. (b) Estimated lifetimes as function of season for ten source regions. (c) Estimated NO$_x$ lifetimes for the selected 50 sources as a function of latitude. (d) Estimated lifetimes in dependence of latitude (black) separated into winter (blue), spring (green), summer (red) and autumn (yellow). All plots are based on TROPOMI NO$_2$ data from 01 March 2018 to 29 February 2020. Sources from southern hemisphere are mirrored in latitude and season. Given uncertainties and error bars are the standard deviation derived by the EMG fitting procedure.

dependence of NO$_x$ lifetimes. This can be used for studies similar to those of Beirle et al. (2019, 2021) where an assumption about the lifetime is necessary to calculate NO$_x$ emissions and can also provide relevant observational constraints for model simulations of NO$_x$ lifetimes.

## 4.4 Weekend effect

The presence of a weekend effect in NO$_2$ on a global scale was first shown by Beirle et al. (2003) with GOME measurements. Anthropogenic activities have their maximum during the week and are reduced during the weekend. Thus, we expect less NO$_x$ emissions in cities at weekends. This behavior should be reflected in a comparison of NO$_x$ emissions on weekdays and weekends. How large the difference between weekends and weekdays is depends on the types of NO$_x$ sources and the different patterns of anthropogenic activity in the source region. To investigate the weekly cycle, the TROPOMI data is separated into

week and weekend days and $NO_x$ emissions and lifetimes were calculated separately. Weekend days can be one or two days and those days can also differ according to religious tradition. For source regions for example in Europe and the United States, weekend days were set to be Saturday and Sunday, for Saudi Arabia weekend days are Friday and Saturday (see also Table A1). Often one weekend day can be different than the other, for example, emissions on Saturdays are often not as low as those on Sundays, even if they are already lower than those from Monday to Friday (Crippa et al., 2020; Goldberg et al., 2021). If the weekend would be limited to one day in the analysis, this would lead to a strengthening of the weekend-to-week ratio for some cities.

Figure 7 shows the weekend-to-weekday ratio, with higher $NO_x$ emissions during weekdays than on the weekend for most of the source regions albeit with rather high variability. For Paris (France), the $NO_x$ emissions are reduced by $40\,\%$ on the weekend, which agrees with Lorente et al. (2019). Chicago shows only a reduction of $16\,\%$ on the weekend versus weekdays which is less than the weekend reductions of $30\,\%$ found in Goldberg et al. (2019, 2021). This could be attributed to the fact that in Goldberg et al. (2019, 2021) Sunday was assumed to be the only weekend day. This results in a more pronounced weekend reduction than with Saturday and Sunday as weekend days as done in this study.

 Source regions not showing any reductions on the weekend are mostly dominated by power plants. Rabigh in Saudi Arabia is a small city with a large gas-fired power plant. Medupi/Matimba are two large coal fired power plants in South Africa. Sarir Field is a large oil field in Libya. Hwange in Zimbabwe is a small town with Zimbabwes biggest power plant. Chelyabinsk (Russia) is a city with nearly 1.2 million inhabitants but also a center of heavy industry and Hassi Messaud in Algeria is an oil refinery town. Source regions with less industry, which are dominated by domestic and transport emissions like Tokio, Brisbane or Madrid on the other hand show large emission reductions on the weekend of $50\,\%$ to $60\,\%$.

Despite showing large $NO_2$ columns, Chinese cities did not show a weekend effect in previous studies (Beirle et al., 2003; Stavrakou et al., 2020). Stavrakou et al. (2020) showed an average weekend-to-weekday ratio in $NO_2$ column over all large Chinese cities of $0.97 \pm 0.02$ using 2005 - 2017 OMI $NO_2$ columns and one year of TROPOMI $NO_2$ columns (May 2018 - April 2019). Ratios of $NO_2$ columns are related but not identical to ratios in $NO_x$ emissions as discussed here, as effects from meteorology and lifetime are not accounted for as well as tropospheric background is contained which is removed in the EMG method by fitting the background. Using the EMG method to investigate the weekly cycle we calculated a weekend-to-weekday ratio of $0.79 \pm 0.3$ in $NO_x$ emissions for Wuhan (China). The ratio includes 27 weekend days and 67 weekdays during the period March 2018 - 22 January 2020. Due to the small number of isolated point sources and because of the limited statistics from using two years of data it was not yet possible to determine weekly cycles for other Chinese cities. The ratio for Wuhan indicates a reduction of $NO_x$ emissions on rest days also in China in the recent years, in contrast to earlier studies showing no such effect. This can be analyzed in more detail as more TROPOMI data become available.

A side effect of the reduced $NO_x$ emissions on weekends might be lower OH levels due to the slower production rate of OH from the $NO + HO_2$ reaction and therefore longer $NO_x$ lifetimes (Stavrakou et al., 2008). However, in our data set the retrieved lifetimes for week and weekend days do not show a clear enhancement of lifetimes on weekends (see Fig. A1). This may possibly be due to insufficient number of data points, required for a statistically significant result and this question should be revisited once a larger TROPOMI data set becomes available.

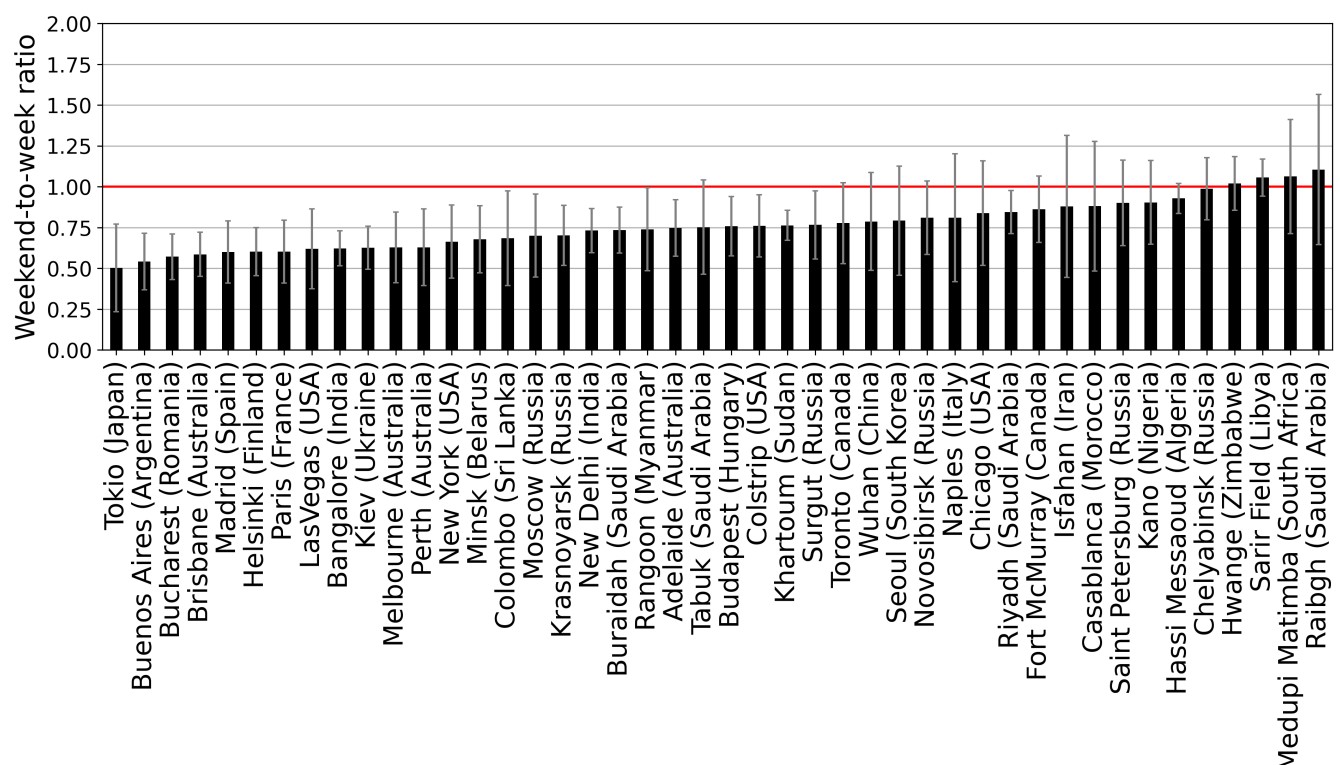

**Figure 7.** Weekend-to-weekday ratio of $NO_x$ emission data (01 March 2018 to 29 February 2020). From left to right in increasing ratio. Data is separated into working and weekend days for 44 source regions. Weekend can be one or two days and those days can also differ according to religious tradition, which is considered. The red line indicates the 1:1 line where weekend and week $NO_x$ emissions are equal, below the line emissions during the week are predominant and above the line weekend emissions are higher. Error bars are the standard deviation derived by the EMG fitting procedure for emission estimates.

## 4.5 Covid-19 effect

In early 2020, several countries took containment measures to prevent the spread of the coronavirus outbreak (Covid-19), which caused reductions in industrial activities and traffic volume. Due to the link of $NO_x$ emissions to human activities, it is possible to investigate the impact of the Covid-19 induced activity reductions with the TROPOMI $NO_2$ column data (Bauwens et al., 2020; Liu et al., 2020; Goldberg et al., 2020). Part of the observed decreases in the $NO_2$ columns may result from effects other than the measures designed to limit transmission of Covid-19, e.g. meteorological variability, seasonal variability
or environmental policy regulations. Consequently, it is problematic to identify a clear Covid-19 effect using only $NO_2$ column amounts. Besides the possibility of using models to separate the Covid-19 effect from other factors (Goldberg et al., 2020), it is also possible to use the EMG method. This approach accounts for wind conditions and $NO_x$ lifetime, which influence the $NO_2$ columns observed for similar $NO_x$ emissions. Since the EMG method can only be used to investigate point sources, the

range of possible study areas is limited. In addition, the analysis of the Covid-19 effect is also much more limited compared to
the rest of the study, because we decided to compare only monthly means from two different years with each other to minimize
influence by meteorology. We focused on the cities Buenos Aires (Argentina), New Delhi (India) and Madrid (Spain), which
are considered to be point sources and have an appropriate number of days with satellite data available during the comparative
periods. The EMG method was used, and monthly means of $NO_x$ emissions from 2019 and 2020 were calculated and compared.
If only a few days per month are available, fit results can be of low quality if conditions on these days are not ideal and statistics
are missing so that, for example, the variation of $NO_x$ emissions on weekdays can also have significant influence. Consequently,
even for the selected cities not all months can be considered.

Figure 8 shows the monthly means from January to November of the calculated $NO_x$ emissions of TROPOMI data for 2019
and 2020 for (a) Buenos Aires (b) New Delhi and (c) Madrid. The same months in 2019 and 2020 are compared, and thus
pre Covid-19 periods with periods of containment measures. For analyses where $NO_X$ emissions from different months are
compared, seasonality of $NO_x$ emissions must be considered. The $NO_x$ emissions retrieved for Buenos Aires are low during
summer from January to March (months 1-3), increase in the winter months with a maximum in July and then decrease again
towards summer. The emissions for New Delhi do not show such a strong seasonality as those for Buenos Aires but in general
are also higher during the winter months January and February (months 1-2). The seasonality is also clearly visible for Madrid,
which must be considered in possible comparisons.

On 20 March 2020, a strong nationwide lockdown began in Argentina and lasted in Buenos Aires for more than seven months,
ending on 8 November. From January to March, $NO_x$ emissions from 2020 are comparable with those from the same months
in 2019. In April 2020, the first complete month in lockdown, the emissions are $57\%$ lower than in April 2019, and also in
May 2020, the emissions are $36\%$ reduced compared to 2019. In June, however, emissions are higher in 2020 than in 2019,
although there has been no major change in the containment measures by the government. A possible explanation is that the
June 2019 emissions are lower than expected from the seasonal cycle comparing to May and July 2019. It is also possible that
June 2020 emissions are unexpectedly high due to a cold winter month and additional emissions from heating. This is also
indicated by ERA5 reanalysis temperature data. They were averaged over the boundary layer in the Buenos Aires target area
for the measurement days used in our study in May, June and July 2019 and 2020. They show that June 2020 was on average
3°C colder than 2019. In comparison, temperatures for May and July 2019 and 2020 are very comparable. The $NO_x$ emissions
in July behave in a manner similar to those in March and April, but have a somewhat smaller decrease of $26\%$ compared to
2019. These reductions are similar to the reductions of $31\%$ and $44\%$ in August and September. In October the $NO_x$ emissions
are almost equal for both years and in November 2020 they are higher than in 2019, possibly due to the end of lockdown on 8
November.

In India, a nationwide strict lockdown started on 24 March 2020. In January the estimated $NO_x$ emissions are almost equal for
both years. The significantly higher emissions in February 2020 compared to 2019 probably cannot be explained by a Covid-19
effect. They are more likely a result of the limited number of clear-sky days for TROPOMI observations, of on average only
five days during February 2019, which will reveal more natural variability. In April 2020, the first complete month in lockdown,
the emissions are $87\%$ lower than in April 2019, and also in May and June the emissions are $54\%$ and $31\%$ lower than in

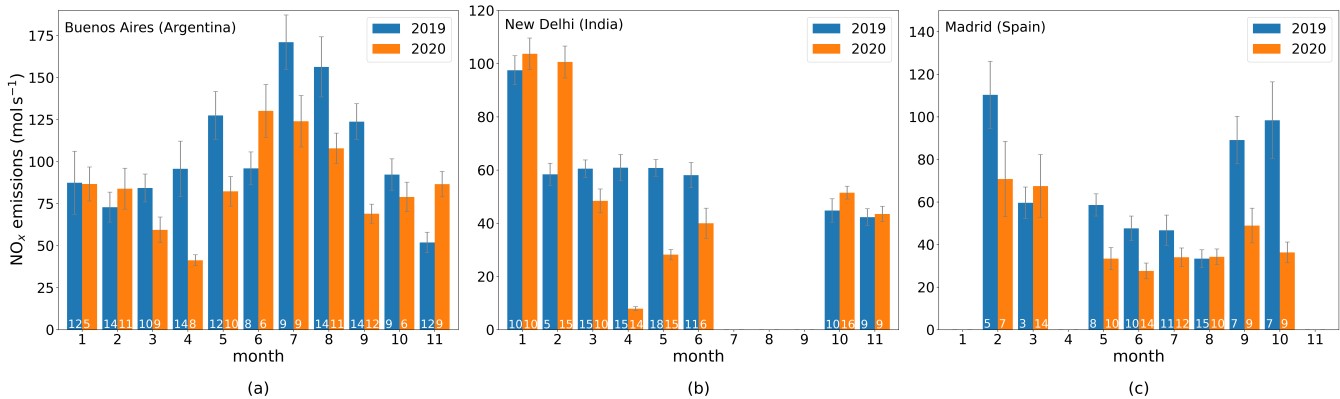

**Figure 8.** Monthly NO$_x$ emissions calculated with the EMG method using TROPOMI NO$_2$ tropospheric column data for 2019 (blue) and 2020 (orange) from January to November for (a) Buenos Aires (Argentina), (b) New Delhi (India) and (c) Madrid (Spain). The numbers in the bars represent the number of available days for the monthly mean. Due to insufficient data availability of less than three days, comparisons are not possible for some months. Error bars are the standard deviation derived by the EMG fitting procedure for emission estimates.

2019. For July to September, comparisons are not possible as less than three days of measurements per month are available
due to cloud coverage and statistic and fit results are not good. In October and November 2020, emissions are almost equal to 2019 levels, suggesting that NO$_x$ emissions are similar to that before the Covid-19 containment measures.

In Europe Madrid was one of the cities strongly impacted by measures to prevent the spread of Covid-19. A strict lockdown was enacted in the middle of March 2020, some restrictions were lifted from middle of April and in May the government followed a plan for easing lockdown restrictions slowly. Due to again rising number of cases, new restrictions started in early
October 2020. Emission comparisons are not possible for January, April and November due to lack of data caused by cloud coverage. The significantly higher emissions in February 2019 compared to 2020 are probably mainly caused by two factors, first a strong synoptic meteorological variability in Europe and second February is also a month with typically persistent cloud cover, resulting in a reduced number of TROPOMI observations which will reveal more natural variability. Similar findings are also described in Bauwens et al. (2020) and Levelt et al. (2021). For the months May and October 2020, when Covid-19
restrictions were in place, the calculated NO$_x$ emissions in Madrid are 43 % respectively 63 % lower than in 2019. This is comparable to results from NO$_2$ column comparisons which showed reduction of around 30 % from the middle of March to early April 2020 relative to 2019 (Bauwens et al., 2020). The larger reductions in October may be due to stricter restrictions resulting in larger emission reductions. However, even with monthly averages, the high variability of NO$_x$ emissions is still not negligible and could also have an influence here.

Despite the shortness of the periods available for analysis, it is possible to investigate short-term variability of NO$_x$ emissions induced by Covid-19 with TROPOMI NO$_2$ data and the EMG method. Strong decreases due to lockdown measures of 57 % in Buenos Aires and 87 % in New Delhi are shown for April 2020 compared to April 2019, as well as a general tendency towards lower emissions in 2020 after the start of the Covid-19 pandemic than 2019. Nevertheless, even with monthly averages, the

high variability of $NO_x$ must still be considered for particular months. These emission estimates account for wind conditions and $NO_x$ lifetime and can therefore give a better estimate of the Covid-19 measures on $NO_x$ emissions than just comparing $NO_2$ column measurements. For some cities and months, the number of days in monthly means is limited due to cloud cover. This is also a problem when comparing monthly $NO_2$ column levels.

## 4.6 Uncertainties

The uncertainties and error bars for emission and lifetime estimates given in this work are based on the standard error (one standard deviation), derived by the fitting procedure and are calculated with error propagation. For emissions this results in:

$$\sigma_E = \frac{\overline{w} \cdot \sigma_{E'}}{N_A} + \frac{E' \cdot \sigma_{\overline{w}}}{N_A} \tag{10}$$

and for lifetimes in:

$$\sigma_\tau = \frac{\sigma_{x_0}}{\overline{w}} + \frac{x_0 \cdot \sigma_{\overline{w}}}{\overline{w}^2} \tag{11}$$

with the emission factor $E'$, the e-folding distance $x_0$, the mean wind speed $\overline{w}$, the Avogadro's constant $N_A$ (see section 1) and the standard deviations $\sigma_{E'}$ of $E'$ and $\sigma_{x_0}$ of $x_0$ derived from the fit and $\sigma_{\overline{w}}$ derived from the wind field in the line density sector. These estimates are based on the fitting uncertainties.

However, the $NO_x$ emissions and lifetimes derived from TROPOMI $NO_2$ data are influenced by additional error sources. The most important contribution directly influencing our estimates is the accuracy of the TROPOMI $NO_2$ tropospheric vertical column itself. This uncertainty is dominated by the accuracy of the tropospheric air mass factor (AMF) and is estimated to be in the order of $30\%$ (Boersma et al., 2004, Bucsela et al., 2013). Recent studies comparing TROPOMI $NO_2$ column with co-located ground based or aircraft measurements reported a low bias for TROPOMI $NO_2$ columns, which is most likely caused by a-priori information such as the surface albedo, cloud-top-height, cloud fraction and the $NO_2$ vertical profile, used for tropospheric AMF calculations. This bias differs for different regions and is more pronounced for regions with larger $NO_2$ columns (Griffin et al., 2019; Ialongo et al., 2020; Judd et al., 2020; Dimitropoulou et al., 2020; Verhoelst et al., 2021). Some studies scaled up the measured $NO_2$ columns with a factor of 1.33 for Paris (Lorente et al., 2019) up to a factor of 1.98 for Germany (Beirle et al., 2019). As this suspected underestimation is not yet fully characterized, and as it is not clear without independent measurements how much the various regions used in our study are affected, we decided not to correct the operational product. Therefore, $NO_x$ emissions derived in this study are systematically biased low. A new operational $NO_2$ product version V01.04., activated after the analysis period of this study on 02 December 2020, implemented first major changes, which led to a substantial increase in the tropospheric $NO_2$ column over polluted areas for scenes with small cloud fractions. Further major updates in the TROPOMI NO2 retrieval algorithm are done in V02.02. released on 1 July 2021. Tropospheric vertical columns are between 10 and $40\%$ larger than the v1.x data depending on the level of pollution and season; the largest impact is found in wintertime at mid- and high-latitudes (van Geffen et al., 2021). A complete mission reprocessing will be performed to get a harmonized data set (Eskes and Eichmann, 2021). The use of the reprocessed data set for this study will increase $NO_x$ emissions and probably affect the seasonality analysis, but not so much the weekend-to-week comparison results.

To calculate $NO_x$ emissions, we applied a conversion of each TROPOMI pixel from $NO_2$ column to $NO_x$ column by assuming that the Leighton photostationary state applies for the polluted air masses. This is more accurate than using a fixed conversion factor as in many earlier studies, especially for our analysis over a large latitudinal range and for different seasons. Nevertheless, the photolysis frequencies have to be parameterized and the temperatures for the rate constant and the ozone concentrations taken from the monthly ERA5 data set interpolated to the S5P overpass time. Thus, the conversion from $NO_2$ to $NO_x$ adds systematic errors in the emission estimates.

In addition, using satellite data introduces a systematic clear-sky bias, because only measurements from nearly cloudless days are used, which favor specific emission patterns. These may differ from those of cloudier and thereby often cooler days. The limitation to nearly cloudless measurements also influence lifetime estimates, which are systematically lower due to higher photolysis rates on cloudless days. Furthermore, the retrieved $NO_x$ emissions and lifetimes are based on measurements in the early afternoon and are therefore systematically biased due to the measurement time. The variability in time can be further analyzed using follow-up sensors on geostationary satellites as for example GEMS, Tempo or Sentinel-4.

The $NO_2$ tropospheric columns are strongly affected by the wind fields. This affects the calculation of $NO_x$ emissions and lifetimes. We filtered the $NO_2$ measurements depending on the corresponding wind speed and only data with wind speeds > $2\,\mathrm{m\,s^{-1}}$ are included in the analysis. As a result of the short lifetime of $NO_2$, the observed $NO_2$ distribution should, in general, be dominated by the wind conditions around the satellite overpass. On days with rapidly changing wind directions around the time of measurement, the spatial patterns and thus the estimates of emissions and lifetimes may be affected. An effect observed for some locations on some days are curved plumes, which for the case of strong curvature leads to a part of the plume being outside of the line density calculation sector and an underestimation of both $NO_x$ emissions and lifetime. This has a large effect when analyzing estimates for single days and is generally larger for longer lifetimes, so especially on winter days. It has a smaller influence on the overall result analyzing a larger average, if not too many days are affected by rapidly changing wind directions. Nevertheless, the wind field is the largest uncertainty (random and systematic) influencing our estimates after the $NO_2$ column itself. Lorente et al. (2019) have modified wind speeds by 20 % and found that emissions changed by 20 %, which demonstrates the strong influence of wind speed on $NO_x$ emissions. However, reliable global wind information is hard to obtain. Beirle et al. (2011) estimated an uncertainty of 30 % for the wind data. The uncertainties due to the chosen wind fields will vary for different source regions. Overall, we assume an uncertainty of 30 %.

To avoid interference from sources of $NO_x$ surrounding the target region, only rather isolated source regions are chosen for the analysis. Since almost no site is perfectly isolated, sectors were defined in which the line density was calculated by integration and fitted with the EMG method to minimize interference between different sources. Due to the rotation of measurements with their corresponding wind direction around the source, the $NO_2$ signal from sources in the surroundings is smeared around the source region in their distance to the source location. To exclude these contaminants, the sector size used for the EMG method has to be chosen carefully. In order to have an adequate amount of data for a robust EMG fit, the sector must first be large enough in both downwind and across-wind directions, but the size is also influenced by other factors. The sector length in wind direction is mainly determined by the influence of other sources but also the spatial extent of the source region itself and wind speeds. A typical size is 300 km, 100 km upwind and 200 km downwind of the source location and is adjusted visually, if

necessary, by inspecting the $NO_2$ distribution and line density. If the influence of surrounding sources is negligible or becomes negligible by adjusting the sector size, the EMG method is robust in variation of the sector size in wind direction. The sector width in across-wind direction is mainly influenced by the geographical extent of the source region. If the sector width is chosen too small, part of the $NO_x$ emissions are outside of the sector due to dilution by wind or due to curved plumes, as described above. This obviously leads to an underestimation of the calculated emissions which acts as an additional apparent loss, leading to the e-folding distance $x_0$ being systematically biased low and also lifetimes, defined as $\tau = x_0 \cdot w^{-1}$, with mean wind speed $w$, are underestimated. Typical sector widths vary between $30\,\mathrm{km}$ and $140\,\mathrm{km}$ and are determined by visually inspecting the $NO_2$ distribution after rotation. Beirle et al. (2019) estimated the uncertainty of $NO_x$ emissions and lifetimes due to sector size to $10\,\%$.

The EMG method is well suited to investigate point sources. In reality, most of the analyzed sources deviate from the assumption of a point source. Isolated sources as the power plant Colstrip in Montana, USA or the Sarir Oil Field in Libya are close to being point sources. In previous studies, the Four Corners and San Juan power plants in New Mexico, USA, which are located $13\,\mathrm{km}$ apart, were investigated as one source using the EMG method (Beirle et al., 2011; Goldberg et al., 2019). With the higher resolution of TROPOMI compared to OMI, it becomes clear that the situation with the two plumes is more complex. This is evidenced by strong irregularities in the line density and fit, indicating that it should not be treated as one point source. In this particular case, re-gridding the TROPOMI data to a coarser resolution would potentially allow an analysis. Other sources such as Tokyo, Moscow or Chicago, all cities with emissions originating from a larger area, are treated as extended point sources but additional uncertainties must be considered. For example, it is not possible to consider a change of the instantaneous $NO_x$ lifetime downwind of the source with the EMG method and estimated lifetimes should be interpreted as an effective mean lifetime (Beirle et al., 2011). This effect is particularly pronounced in spatially extended source areas and can lead to low biased lifetimes.

In summary, the total uncertainty of $NO_x$ emissions and lifetimes derived from TROPOMI $NO_2$ data is influenced in addition to the 1-sigma uncertainties, derived by the fitting procedure, by many different error sources, which are mainly systematic and lead to an overall low bias of the derived emissions. Dominated by the systematic errors in the TROPOMI $NO_2$ tropospheric column itself ($30\,\%$ - $50\,\%$) and in the wind field ($30\,\%$). The sector size can lead to low biased emissions and lifetimes and is estimated to introduce an uncertainty of $10\,\%$. Further uncertainties due to the measurement time, the clear-sky bias and assumptions about point sources tend to have low bias in the lifetimes and emissions. More analysis is needed to resolve these issues. The error contributions summed in quadrature result in an overall uncertainty estimate for lifetime and emissions in the range $43\,\%$ - $62\,\%$.

## 5 Conclusions

In this study, we present investigations of the variability of $NO_x$ emissions and lifetimes estimated from Sentinel-5P TROPOMI observations of selected $NO_x$ source areas around the world. Similar to earlier studies, we combine TROPOMI $NO_2$ tropospheric vertical column data with wind information from ECMWF ERA5 reanalysis for the exponentially modified Gaussian,

EMG, method. TROPOMI measurements with their high spatial resolution and high signal to noise ratio enable the detailed analysis of $NO_x$ emissions and lifetimes using only two years of data. Investigating small emission sources not analyzed before and also monitoring the variability on a short-term temporal basis has been possible. Here, a total of 50 $NO_x$ sources from different regions, located between the equator and 61° latitude have been investigated. Despite efforts to select a broad range of regions, well distributed around the globe and a mix of different sources, predominantly cities, isolated power plants, industrial regions, oil fields and regions with a mix of sources, some selection bias could have been introduced, for example, by not analyzing regions with low wind speeds, mostly cloudy conditions or areas in regions with many emission sources and higher background $NO_2$ levels.

Comparisons to other studies using TROPOMI data and similar methods for emission estimates show small differences and in general good agreement taking into account the differences of the analyses. The emission estimates are compared to the EDGAR (v.5.0 2015) emission inventory. For most of the source regions the emissions are higher than those estimated with the EMG method. These differences are particularly strong for the source regions with the highest emissions. As the operational TROPOMI tropospheric $NO_2$ product has been reported to be low in comparison to independent measurements, part of the apparent overestimation by EDGAR could be related to a TROPOMI low bias. On the other hand, $NO_x$ emission trends over the last five years are likely to also have an impact. EPA CEMS data for the Colstrip power plant shows that power plant emissions can vary strongly over time. The EMG method applied to TROPOMI data, at least for this power plant, reproduces the temporal variability seen in the CEMS data reasonably well, and does not show a significant seasonal bias. Consequently, we consider that the method also give reasonable results analyzing winter data. However, the TROPOMI based $NO_x$ emissions are lower by about a factor of two for this power plant as compared to the CEMS data, in agreement with earlier findings for other power plant sources.

The seasonal separation of the emission estimates indicates that the highest emissions occur during winter especially for source regions at higher latitudes. Whereas in hot desert climate the highest emissions occur in summer. This is best explained by the different contributions to $NO_x$ emissions depending on source region, which are typically dominated by domestic heating in winter or, air conditioning in hot summer months, depending on the climatic conditions of the source region. In some respects our results differ from those in previous studies using emission inventories. These studies have analyzed emissions for countries, where generally weaker seasonality is found. However, this probably does not reflect the specific situation in cities, where the composition of sources is expected to be different. Further detailed investigations and comparisons would be helpful to understand the discrepancies between emission estimates using emission inventories and the EMG method with satellite data. Generally, it should be considered that the uncertainties for emission estimates from the EMG method can be large and especially the analysis of winter data has not been performed often and is influenced even more by uncertainties.

The investigation of the seasonal and latitudinal dependence of the $NO_x$ lifetime shows an increase in lifetime from two to six hours for increasing latitudes, but only a weak seasonal dependence with longer lifetimes in winter than in summer. Larger differences are expected based on modelling studies. The weak seasonal dependence found in this study could be explained by not yet sufficient statistics, a clear-sky bias and also the midday observation time of TROPOMI, which all lead to more balanced lifetimes in summer and winter. Uncertainties which tend to have a low bias in lifetime estimates, especially in winter

due to longer lifetimes and the aforementioned points, should be considered.

Separating $NO_x$ emission estimates into working and weekend days indicates that for most $NO_x$ source regions, emissions are higher during weekdays than on the weekend, albeit with rather high variability in the weekend-to-weekday ratios. Only source regions, which are dominated by power plants or industry, do not show any reductions in $NO_x$ emissions on the weekend. The largest emission reductions on weekends are found for source regions with little industry, which are dominated by city emissions mostly from traffic. Our results indicate reductions of $NO_x$ emissions for Wuhan (China) during weekends, in contrast

to earlier studies showing no such effect in China. This study can be extended to other Chinese cities, as more high resolution tropospheric $NO_2$ columns become available from TROPOMI and GEMS in the next years. Separately calculated lifetimes for weekend and working days do not show longer lifetimes during weekends. This result also requires further investigation.

A short-term reduction in emissions attributable to the measures introduced to limit the spread of Covid-19 infections was found by comparing $NO_x$ emissions estimated with the EMG method. Since the method accounts for wind conditions and

725 $NO_x$ lifetimes, it gives a better estimate on the impact of the Covid-19 measures on $NO_x$ emissions than comparisons of tropospheric $NO_2$ column measurements as done in other studies. Strong $NO_x$ emission reductions during the first lockdown phase and a general tendency to lower emissions in 2020 than in 2019 are shown for Buenos Aires, New Delhi and Madrid.

We conclude that the EMG method in combination with the high spatial resolution TROPOMI $NO_2$ tropospheric column measurements allows us to investigate the high variability of $NO_x$ emissions and lifetimes on a global scale and on short time

frames. Depending on the goal of the analysis, already a few days of measurements can be sufficient; for seasonal studies, depending on the local meteorological conditions, one to two seasons are sufficient, and it is not anymore necessary to average over several years. The ability to estimate emissions over short periods will also allow policy makers to evaluate $NO_x$ emission regulations better and more quickly. The presented high variability should be further investigated using follow-up sensors on geostationary satellites as for example GEMS, Tempo or Sentinel-4, which have the potential to investigate in addition the

diurnal variability.

# Appendix A

**Table A1.** $NO_x$ source regions sorted with increasing latitude with mean $NO_x$ lifetime and emissions, wind speed, available days, season flag and weekly cycle flag (x: possible, –: not possible). Errors are the standard deviation derived by the EMG fitting procedure.

| Source region | Latitude (degree) | Longitude (degree) | $NO_x$ lifetime (hours) | $NO_x$ emissions (mol s$^{-1}$) | wind speed (m s$^{-1}$) | days | seasons | weekdays |
|---|---|---|---|---|---|---|---|---|
| Singapore (Singapore) | 1.30 | 103.69 | $2.15 \pm 0.28$ | $94.5 \pm 11.5$ | 5.44 | 24 | – | – |
| Lagos (Nigeria) | 6.55 | 3.40 | $2.37 \pm 0.49$ | $30.9 \pm 6.1$ | 4.31 | 29 | – | – |
| Colombo (Sri Lanka) | 6.93 | 79.85 | $1.22 \pm 0.12$ | $22.8 \pm 2.3$ | 6.17 | 61 | x | Sat, Sun |
| Kano (Nigeria) | 11.98 | 8.51 | $2.55 \pm 0.19$ | $5.1 \pm 0.3$ | 4.65 | 317 | x | Sat, Sun |
| Bangalore (India) | 12.98 | 77.59 | $1.93 \pm 0.08$ | $20.0 \pm 0.8$ | 4.66 | 155 | – | Sun |
| Khartoum (Sudan) | 15.58 | 32.52 | $1.27 \pm 0.04$ | $16.9 \pm 0.4$ | 5.85 | 494 | x | Fri, Sat |
| Rangoon (Myanmar) | 16.78 | 96.15 | $2.18 \pm 0.2$ | $14.7 \pm 1.2$ | 3.64 | 127 | – | Sat, Sun |
| Hwange (Zimbabwe) | -18.38 | 26.47 | $1.63 \pm 0.06$ | $5.0 \pm 0.1$ | 4.72 | 254 | x | Sat, Sun |
| BeloHorizonte (Brazil) | -19.91 | -43.98 | $4.09 \pm 0.34$ | $10.3 \pm 0.7$ | 3.76 | 64 | – | – |
| Guadalajara (Mexico) | 20.66 | -103.34 | $1.23 \pm 0.14$ | $35.5 \pm 4.1$ | 5.00 | 175 | – | – |
| Raibgh (Saudi Arabia) | 22.69 | 39.03 | $1.22 \pm 0.13$ | $69.1 \pm 7.2$ | 6.32 | 311 | x | Fri, Sat |
| Medupi Matimba (South Africa) | -23.68 | 27.58 | $3.68 \pm 0.32$ | $55.5 \pm 4.5$ | 4.81 | 211 | x | Sat, Sun |
| Riyadh (Saudi Arabia) | 24.65 | 46.71 | $2.91 \pm 0.11$ | $186.1 \pm 6.8$ | 6.32 | 360 | x | Fri, Sat |
| Buraidah (Saudi Arabia) | 26.20 | 43.99 | $2.57 \pm 0.14$ | $25.4 \pm 1.1$ | 5.74 | 418 | x | Fri, Sat |
| Brisbane (Australia) | -27.47 | 153.03 | $2.31 \pm 0.12$ | $24.1 \pm 1.0$ | 5.66 | 166 | x | Sat, Sun |
| Sarir Field (Libya) | 27.55 | 21.63 | $1.88 \pm 0.04$ | $5.5 \pm 0.1$ | 5.84 | 442 | x | Fri, Sat |
| Tabuk (Saudi Arabia) | 28.48 | 36.52 | $1.34 \pm 0.14$ | $24.3 \pm 2.3$ | 5.47 | 342 | x | Fri, Sat |
| New Delhi (India) | 28.62 | 77.22 | $3.05 \pm 0.12$ | $69.8 \pm 2.5$ | 5.09 | 204 | x | Sun |
| Wuhan (China) | 30.57 | 114.28 | $2.94 \pm 0.3$ | $115.1 \pm 10.7$ | 5.10 | 94 | x | Sat, Sun |
| Hassi Messaoud (Algeria) | 31.70 | 6.05 | $2.57 \pm 0.06$ | $8.2 \pm 0.2$ | 6.17 | 394 | – | Fri, Sat |
| Perth (Australia) | -31.95 | 115.85 | $2.43 \pm 0.24$ | $16.2 \pm 1.5$ | 6.35 | 232 | x | Sat, Sun |
| Isfahan (Iran) | 32.64 | 51.67 | $2.08 \pm 0.24$ | $118.8 \pm 12.6$ | 5.38 | 178 | – | Fri |
| Casablanca (Morocco) | 33.59 | -7.61 | $2.98 \pm 0.35$ | $18.6 \pm 2.1$ | 4.93 | 265 | x | Sat, Sun |
| Xi'an (China) | 34.27 | 108.94 | $3.37 \pm 0.49$ | $164.2 \pm 23.5$ | 4.90 | 33 | – | – |
| Buenos Aires (Argentina) | -34.50 | -58.80 | $3.51 \pm 0.27$ | $80.2 \pm 5.8$ | 6.45 | 207 | x | Sat, Sun |
| Adelaide (Australia) | -34.92 | 138.60 | $3.11 \pm 0.2$ | $20.5 \pm 1.1$ | 6.97 | 175 | x | Sat, Sun |
| Tokio (Japan) | 35.68 | 139.77 | $3.5 \pm 0.42$ | $212.0 \pm 24.5$ | 7.36 | 39 | – | Sat, Sun |
| LasVegas (USA) | 36.16 | -115.19 | $1.97 \pm 0.2$ | $21.2 \pm 2.0$ | 6.23 | 235 | x | Sat, Sun |
| Seoul (South Korea) | 37.60 | 127.00 | $3.77 \pm 0.4$ | $272.9 \pm 28.2$ | 6.28 | 124 | – | Sat, Sun |

**Table A2.** Continuation of table A1

| Source region | Latitude (degree) | Longitude (degree) | $NO_x$ lifetime (hours) | $NO_x$ emissions (mol s$^{-1}$) | wind speed (m s$^{-1}$) | days | seasons | weekdays |
|---|---|---|---|---|---|---|---|---|
| Melbourne (Australia) | -37.80 | 144.95 | 2.53 ± 0.24 | 46.8 ± 3.7 | 7.19 | 137 | x | Sat, Sun |
| Madrid (Spain) | 40.41 | -3.70 | 2.03 ± 0.15 | 60.2 ± 4.3 | 5.43 | 155 | x | Sat, Sun |
| New York (USA) | 40.71 | -74.01 | 4.35 ± 0.37 | 101.0 ± 8.3 | 7.21 | 133 | x | Sat, Sun |
| Naples (Italy) | 40.83 | 14.25 | 2.26 ± 0.28 | 28.6 ± 3.4 | 6.17 | 109 | x | Sat, Sun |
| Barcelona (Spain) | 41.40 | 2.17 | 3.41 ± 0.63 | 36.7 ± 6.1 | 6.94 | 52 | x | – |
| Chicago (USA) | 41.80 | -87.80 | 5.65 ± 0.52 | 82.9 ± 7.4 | 7.26 | 83 | x | Sat, Sun |
| Toronto (Canada) | 43.66 | -79.38 | 3.16 ± 0.24 | 53.8 ± 3.5 | 6.93 | 69 | – | Sat, Sun |
| Bucharest (Romania) | 44.43 | 26.10 | 3.2 ± 0.18 | 8.1 ± 0.4 | 5.20 | 142 | x | Sat, Sun |
| Colstrip (USA) | 45.88 | -106.61 | 2.58 ± 0.15 | 4.6 ± 0.3 | 6.55 | 165 | x | Sat, Sun |
| Budapest (Hungary) | 47.50 | 19.05 | 2.32 ± 0.13 | 17.5 ± 0.9 | 5.60 | 143 | x | Sat, Sun |
| Paris (France) | 48.86 | 2.35 | 3.79 ± 0.31 | 56.2 ± 4.4 | 5.99 | 116 | x | Sat, Sun |
| Kiev (Ukraine) | 50.45 | 30.50 | 3.11 ± 0.18 | 24.3 ± 1.2 | 6.03 | 143 | x | Sat, Sun |
| Minsk (Belarus) | 53.90 | 27.55 | 6.69 ± 0.59 | 13.8 ± 0.7 | 6.92 | 128 | – | Sat, Sun |
| Novosibirsk (Russia) | 55.15 | 82.98 | 4.88 ± 0.34 | 25.9 ± 1.7 | 6.52 | 105 | x | Sat, Sun |
| Chelyabinsk (Russia) | 55.21 | 61.44 | 3.84 ± 0.23 | 22.6 ± 0.9 | 6.70 | 104 | x | Sat, Sun |
| Moscow (Russia) | 55.95 | 37.62 | 5.96 ± 0.5 | 123.6 ± 9.9 | 6.66 | 70 | x | Sat, Sun |
| Krasnoyarsk (Russia) | 56.10 | 92.93 | 4.88 ± 0.27 | 17.2 ± 0.9 | 5.78 | 96 | x | Sat, Sun |
| Fort McMurray (Canada) | 57.17 | -111.59 | 5.58 ± 0.31 | 21.4 ± 1.1 | 6.25 | 93 | x | Sat, Sun |
| Saint Petersburg (Russia) | 59.95 | 30.40 | 4.06 ± 0.27 | 39.0 ± 2.5 | 6.69 | 75 | – | Sat, Sun |
| Helsinki (Finland) | 60.29 | 24.96 | 8.23 ± 0.44 | 22.4 ± 1.1 | 7.27 | 87 | – | Sat, Sun |
| Surgut (Russia) | 61.25 | 73.43 | 5.2 ± 0.37 | 10.0 ± 0.6 | 6.83 | 70 | – | Sat, Sun |

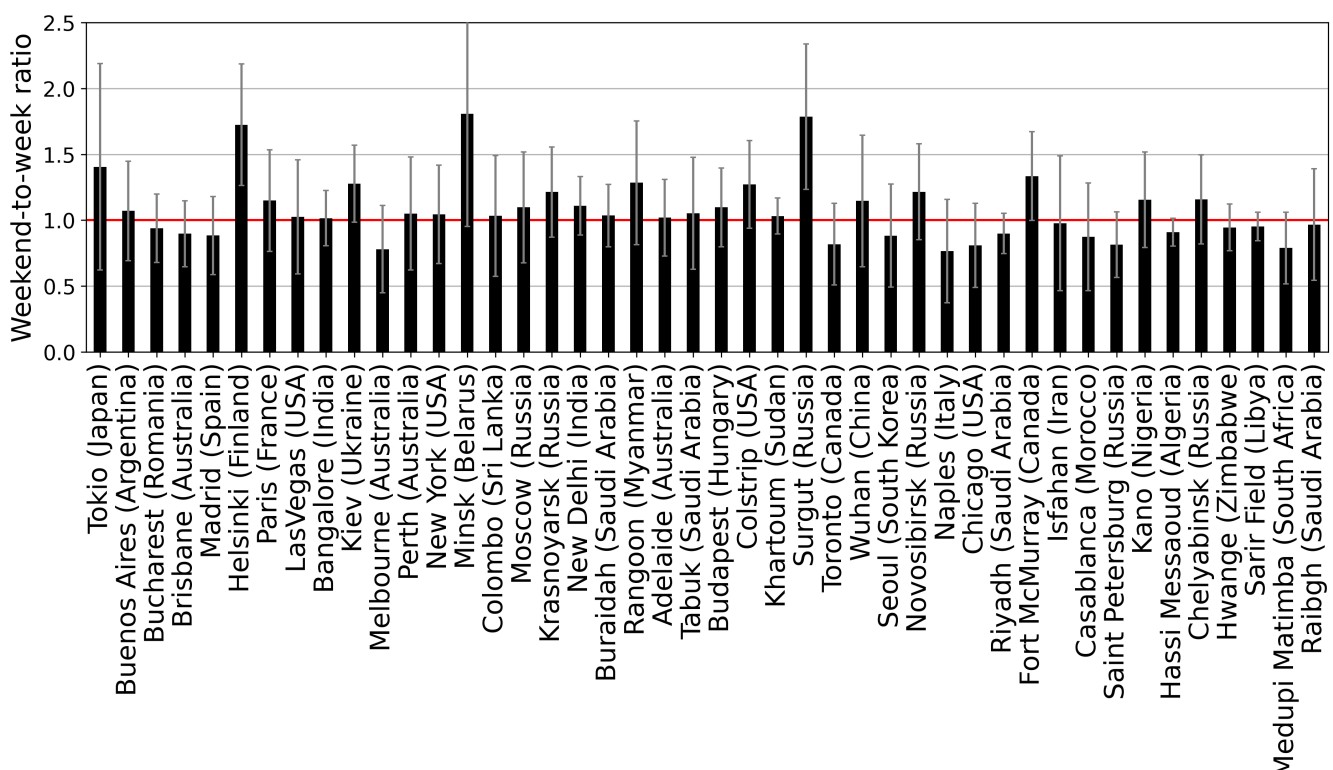

**Figure A1.** Weekend-to-weekday ratio of $NO_x$ lifetimes (01 March 2018 to 29 February 2020). From left to right in increasing weekend-to-weekday $NO_x$ emission ratio (see Fig. 7). Data is separated into working and weekend days for 44 source regions. Weekend can be one or two days and those days can also differ according to religious tradition, which is considered. The red line indicates the 1:1 line where weekend and weekday lifetimes are equal, below the line weekday lifetimes are longer and over the line weekend lifetimes are longer. Error bars are the standard deviation derived by the EMG fitting procedure for lifetime estimates

*Data availability.* TROPOMI data from July 2018 onwards is freely available via https://s5phub.copernicus.eu/. The wind, ozone, temperature and boundary layer height data from the ERA5 reanalysis are freely available from the Copernicus Climate Change (C3S) climate data store (CDS) (http://doi.org/10.24381/cds.adbb2d47). EDGAR v5.0 Global Air Pollutant Emissions are available by https://edgar.jrc.ec.
europa.eu/overview.php?v=50_AP. CEMS data from power plants can be downloaded here: https://ampd.epa.gov/ampd/.

*Author contributions.* KL, AR and JPB designed the study. KL performed the analysis and wrote the paper with contributions from JPB and AR.

*Competing interests.* The authors declare that they have no conflict of interest.

*Acknowledgements.* We gratefully acknowledge the German Aerospace Center (DLR) Bonn for funding the MAXGRAD project – project
no. 50 EE 1709A. Copernicus Sentinel-5P level 2 $NO_2$ data is used in this study. Sentinel-5 Precursor is a European Space Agency (ESA) mission on behalf of the European Commission (EC). The TROPOMI pay-load is a joint development by ESA and the Netherlands Space Office (NSO). The Sentinel-5 Precursor ground-segment development has been funded by ESA and with national contributions from the Netherlands, Germany, Belgium, and UK. Meteorological data is provided by ECMWF ERA5. EDGAR v5.0 Global Air Pollutant Emissions are provided by https://edgar.jrc.ec.europa.eu/overview.php?v=50_AP, DOI: https://data.europa.eu/doi/10.2904/JRC_DATASET_EDGAR.

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
