# Peer review of "Variability of nitrogen oxide emission fluxes and lifetimes estimated from Sentinel-5P TROPOMI observations"

_Atmospheric Chemistry and Physics, 2021_

## Referee Comment (RC2)

**Review "Variability of nitrogen oxide emission fluxes and lifetimes estimated from Sentinel-5P TROPOMI observations", by Lange et al.**

The study by Lange et al. applies the EMG method to TROPOMI NO2 observations to estimate NOx emissions and lifetimes in various urban areas and sources. They study in detail the variability, seasonality, trends and geographical dependence of the NOx emissions and lifetimes. The method is robust and has been applied to satellite data in the past, with the added value of detailed analysis and other sources being analyzed in this study. The paper is well written and well structured, with quality figures supporting the methods and discussion provided in the text.

Although results and conclusions are presented in a clear way, there are some points that need to be addressed to make the conclusions more substantial and robust. So I would recommend for publication once the comments below are addressed by the authors.

**General comments**

In general, the introduction could be improved. Currently the concepts are mixed, and it is not to the point. After reading it, it is not clear what the study is going to be about, and what the research adds to the current literature that is discussed is lacking. It starts with some chemistry, then lifetime, emission estimates, seasonal, day to day, trends, then covid and then back to lifetimes to then introduce the satellite measurements that have already been mentioned at different points in the text before. In the last paragraph you explain what you do in the study but still go back to literature, so not to the point. I think the text and ideas are there, but somewhat disorganized.

Sect. 4.5 and the discussion on the Covid effects on NOx emissions is vague. You argue that EMG method accounts for meteorology and other possible effects on NOx emissions, but the arguments do not hold equally for the three cities that are discussed and different months. Only months where results are supported by the covid hypothesis are brought forward, which makes conclusions less solid. Many cities have been analysed in literature to study covid effects on air pollution, so the fact that only three are presented here is also a sign that the method does not work in other cities. If this is not the case, then this should also be stated in the manuscript.

The results discussed in other sections may be affected by Covid, but there is no mention to that. How are the summer to winter ratios affected by covid? How is the comparison to EDGAR affected by the supposedly lower emissions when restrictions due to covid where active?

Sect. 4.6: The 43-63% uncertainties are higher than the 1-sigma uncertainty provided by the EMG method (e.g. Table 1 uncertainties are ~ 5 %). The high uncertainties need to be reflected in the uncertainty values you provide for the emissions and lifetimes, and discussed in this section.

**Specific comments**

Introduction

Line 101: Is not only urban areas that are the target of the study, right? Same at the beginning of Sect. 5.

Why you do not discuss Beirle et al. (2021) in the Introduction? Is it because those are only point sources?

Data

Line 154 when referring to Fig. 1: "The red circles mark regions with higher NO2 than their surroundings and are analyzed in this study." Is this the reason why you choose these regions? Now it reads as it would be like that, but there are many more places with "higher NO2 than the surroundings".

Wind data and ozone mixing ratio: one is interpolated to the exact overpass time and the other to "typical mean early afternoon overpass time". Why this difference?

Line 191: were "new sources" found in Georgoulias et al. (2019)? Or was it positive trends as well as negative ones, that also reversed during the last decades? Could you use the trends from 2015-2017 from this study to extrapolate EDGAR inventory?

Sect. 3.1.: why you do not choose any region in Australia? Southern Hemisphere is underrepresented in your selection.

Pleases mention here the > 2m/s applied to the overall filtering criteria.

Have you looked at boundary layer height information? 100 m is probably too low, so you might introduce a systematic error here.

Results and discussion

Sect. 4.1 & Sect. 4.2: How do your summer estimates agree with Goldberg et al. (2019)? Are there any temperature anomalies in the areas you study that could point to an over/under estimation of the emission inventories based on the threshold temperatures assumed for heating/air conditioning?

Sect. 4.3: Lifetimes need to be validated in order to discard unrealistic effects on the NOx emissions and ratios shown in Sect. 4.2. Also, in view of the last paragraph of Sect .4.4. Or at least make clear that the uncertainties associated to the lifetimes is much higher than 1-sigma given by the method, which results in important uncertainties in the emission estimates.

Sect.4.4: line 420: what if you only take into account summer data as in Goldberg et al. 2019?

Sect. 4.5: line 469: this can be known by looking at meteorology anomalies in this period.

Line 475: two years cannot be considered a "trend". Has the economy changed so much that is noticeable in emissions?

Line 483: This would indicate that the lockdown was much stricter in May compared to October, so these estimations might be affected by other factors rather than covid, which could point to the lifetime that is estimated being too low.
What is the explanation for higher emissions in Feb. 2020 compared to 2019?

**Technical corrections**

Throughout the manuscript, please modify "nitrogen oxide" by "nitrogen oxides" and "emission" by "emissions" when necessary. NOx is plural, so NOx are emitted (e.g. page 1, line 15).

Please define acronyms first time they are mentioned (e.g. abstract lines 1-2 TROPOMI, NOx)

Abstract: "is ECMWF ERA 5" relevant? Just "wind data" could be sufficient, as there is no need to go into details in the abstract.

Line 90: "but analyses with GEOS-Chem…": reader does not necessarily know what GEOS-Chem is, so please specify.

Line 92: do you mean "measurements of NO2" instead of "measurements of NOx"?

Line 116: "(" missing

Line 152: "removes part of the scenes" -> "removes the scenes"

Line 215: facilities - > facilitates

---

## Author Comment (AC1)

Comments from anonymous Referee #1:

We would like to thank the reviewer for his/her helpful comments. We hope that we could address all questions and unclear points satisfactorily.

In the course of the revision, we have made the following important changes to the analysis: We updated our wind and ozone data and are now using ERA5 reanalysis data averaged about the boundary layer instead of a fixed height to include possible seasonality in these input data. This led to changes in all our estimates but not in any fundamental way. We also applied our method to a power plant for which hourly NOx emissions are reported, and analyzed the temporal variability over the course of the year in more detail. From this comparison we conclude that the EMG method applied to TROPOMI data at least for this power plant can reproduce the temporal variability reasonably well and does not show a clear seasonal bias. To get a better representation of source areas from Southern Hemisphere, we included five new target areas. The results fit well into the overall result of the study.

Legend: Referee comments in black, author comments in blue

This manuscript estimates the NOx emissions from cities and large sources around the globe using TROPOMI NO2 satellite data. I appreciate that the manuscript is thorough and well-written.

I am concerned about the derived NO2 lifetimes by season and month. The magnitude of the seasonal cycle of emissions reported here are in disagreement (and sometimes strong disagreement) with previous literature. Please see Crippa et al., 2020 which shows that CO2 emissions (which can be used as a rough surrogate for NOx in this instance) don't have a very strong seasonal cycle. Using data reported in Crippa et al., 2020… CO2 emissions are maybe 10% lower in summer in the US and China as compared to winter. In Spring and Fall, CO2 emissions are lowest, perhaps 15-20% lower than the winter peak. The summer-to-winter ratios reported in this manuscript, such as ~0.5 in New York City, ~0.4 in Chicago, ~0.25 in Wuhan are simply not reasonable. Even in Europe, Crippa et al., report ratios of ~0.8, and the values reported here are significantly lower.

My guess is that the EMG method is having trouble discerning the true NO2 lifetime during winter. This is a known issue and is why previous literature focus on summer time emissions (Lu et al., 2015; Goldberg et al., 2019). To prove whether your method is reasonable during the winter, I suggest two potential strategies: 1.) You can apply the method to a model simulation that has data during both the winter and summer; since the emissions in the model are known, the method should hopefully reproduce the seasonal cycle of the emissions that are input into the model (whatever they are). 2.) You can apply the method to power plants which report their NOx emissions for all seasons such as all the large power generation facilities in the US. This will hopefully provide insight on whether the seasonal effects on the NO2 lifetime and therefore NOx emissions are being correctly accounted for.

We understand the concerns of the Reviewer about the strong seasonality seen for several cities that also was a surprise to us. We checked the literature from Crippa et al. (2020). Especially Figure 7, showing a time series of monthly fossil $CO_2$ emissions by sector in the world, shows a strong variability over the course of the year. This is a global picture but driven by the top $CO_2$ emitting countries located in the Northern Hemisphere (China, USA, Europe,

Russia). The annual variability is strongly influenced by the power generation and residential combustion sectors, which are showing lower emissions from May to August and higher emission during winter. The residential sector is showing emissions more than three times higher during colder months than during summer. Figure 8 shows more regional results, showing summer to winter ratios from approximately 0.6 for Russia, 0.7 for Europe and 0.85 for the USA, driven by combustion of fuels in the power and residential sectors during cold months. Still, these are very broad areas and it could be expected that results differ on smaller scales (in cities) due to composition of emission sources, policy regulations, and people's behavior.

Nevertheless, we also see the point that the deviations between the seasonality from Crippa et al. (2020) and our results are large and should therefore be questioned. We therefore followed the referee's suggestion and applied our method to a power plant for which hourly NOx emissions are reported, and analyzed the temporal variability over the course of the year in more detail to get more confidence in the method especially during different seasons.

[Figure]

Figure 1: Comparison of CEMS NOx data for the Colstrip (Montana, USA) power plant with emissions estimated by the EMG method, based on TROPOMI data from February 2018 to March 2020. CEMS data are daily mean values, averaged between 11-14 local time, shown in blue and filtered for days with TROPOMI measurements shown in green. Means over periods, marked by the dashed black lines, are shown by the green lines for the CEMS data and red lines for the TROPOMI EMG estimates.

[Figure]

Figure 2: Scatter plot of the CEMS NOx measured emission vs S5P EMG NOx estimates for seven time periods of a range between two to five months during February 2018 to March 2020. The 1:1 line is indicated with the grey dashed line. The solid black line indicates the linear regression, with a slope of 0.34 and a correlation of 0.81.

From this comparison we conclude, that a) power plant emissions can vary strongly over time which was also confirmed by a check of power generation reported by German power plants, b) that the EMG method applied to S5P data at least for this power plant can reproduce the temporal variability reasonably well and does not show a clear seasonal bias, c) that the S5P based $NO_2$ emissions can be lower by about a factor of 2 for power plants as already discussed in previous publications (Beirle et al., 2019, supplement) and usually explained by a mismatch of the model based a priori profile used and the power plant plume which is not well mixed in the proximity of the source.

With the aforementioned said, I think that Sections 4.1 and 4.4 are useful advancements to the literature. I recommend that Sections 4.2, 4.3 and 4.5 are excluded, and included in a follow-up manuscript that more rigorously evaluates the top-down NO2 lifetime by season.

Since we could reproduce the temporal variability of the NOx emissions of the power plant reported by EPA-CEMS in a satisfying way and also regardless of the season, we decided to keep the Sections 4.2, 4.3 and 4.5 where seasonality is analyzed and add the new data to make things a little clearer.

In response to the reviewer's comment, in addition to the new section, in which we show that the method can reproduce temporal variability over the course of the year, we also added comments in which we highlight potential problems when looking at seasonality and included remarks at several corresponding passages throughout the text. Furthermore, we updated our wind and ozone data and are now using averages about the boundary layer instead of a fixed height to include possible seasonality in these input data. This lead to changes in all our estimates but not in any fundamental way.

All suggestions:

Line 39: in-homogeneous —> heterogeneous

Changed as suggested.

Line 43: Missing comma between winter and residential

Added a comma.

Line 62: states —> countries

Done.

Line 147: Availability of data in March and April?

We added a sentence that the data before 30 April is only available on the S5P Copernicus Expert Hub and therefore not publicly accessible.

Figure 1: Suggestion to remove lat/lon lines on the plot. The lines obscures a few cities.

Lat/lon grid is removed from the plot, which gives more visibility for some sources. Some are still partially obscured by coastlines and borders, which cannot be changed without losing information about country affiliation.

Line 163: Wind direction likely similar between heights, but wind speed can be different, especially if comparing near surface to 1000 m. Maybe something to note.

We followed the suggestion and changed our wind data from 100m to wind speed and direction averaged over the boundary layer, extracted from ERA5 re-analysis model level data. Therefore, we adapted the section about the wind data, and included that the chosen height of wind data is not only critical for wind speed but also for wind direction.

Line 163: Might also want to mentioned ease of use of 100-m wind speed since it is a standard variable output in the ERA5 re-analysis.

This is true, that is why we had chosen the 100-m wind data in the beginning. To exclude the possibility that the strong seasonality in the emissions is due to a lack of seasonality in the wind data (seasonal variability of boundary layer height -> higher/lower wind speed), we decided to use an average now over the boundary layer instead of the fixed height of 100 m.

Line 172: Can you clarify? Do you mean every three hours (0Z, 3Z, etc.)? I originally interpreted three hourly estimates to mean three different outputs each hour.

Thank you for pointing out this unclear formulation. It was meant that every three hours a value is available.

As part of the update to the new wind data, we also updated the ozone data which we now also derive from the ERA5 reanalysis data which have a better spatial resolution and much better temporal resolution. Ozone data are averaged over the boundary layer just as wind data and thereby provide a better representation of seasonal variability.

We now use hourly ozone volume mixing ratios with a horizontal resolution of 0.25° in model levels averaged over the boundary layer and interpolated to the TROPOMI overpass time and oversampled to the same 0.01° resolution as the TROPOMI data.

Line 173: Presumably the interpolation is location specific (Europe may use 12Z model data but China may use 3Z for example). Please clarify.

Yes, that is right, the interpolation is location specific. We updated the section about the ozone data and added that the ozone data are interpolated to the TROPOMI overpass time.

Line 207: earth —> globe

Changed.

Line 213: Modify "low cloud coverage" to "few clouds". "Low cloud" could be interpreted to mean low in the atmosphere.

We changed it accordingly in the text.

Line 318: Verhoelst et al., 2021 and Judd et al. 2020, which you already cite, show a low bias in polluted areas of ~20-30%. I see no problem with using this to scale the final estimates by the value, while also noting the large uncertainty in this conversion factor.

We chose not to scale our results due to the difficulty of determining which region needs scaling and which would be the appropriate factor. Different factors are discussed in the literature and due to the different regions and sources discussed, different factors are likely to be needed. However, we now point out in the corresponding sections that we know about the low bias and also that it may be reduced with the reprocessed TROPOMI data version, which is unfortunately not available yet.

Line 336: Early studies, published in 2018 & 2019 used an algorithm that has since been re-processed. Also there was a different rotation in Goldberg et al., 2019; it was done manually.

We checked the Goldberg et al. (2019), which we used for comparison to our results and found that data of "operational v1 varies from v1.00.01 to v1.01.00" were used, these are the same versions used also in our analysis.

We have added the point about the additional manual rotation to the text.

Line 350: I am not convinced that these ratios should be this small. Traffic pollution and industrial/manufacturing pollution, is relatively constant year-round and represents a large fraction of emissions. If anything traffic and industrial emissions would be biased to be larger in the summer (traffic peaks in July and lowest in February: https://www.fhwa.dot.gov/policyinformation/travel_monitoring/tvt.cfm). NO2 lifetime, in theory, should be varying much more than you found. So I'm thinking the summer/winter ratios are an artifact of an erroneous lifetime fit. See Zheng 2018 as an example. They report that residential emissions in China (the sector causing the most intraseasonal variation in emissions) is ~5% of Chinese emissions. Even if this percentage is off by a factor of two and

power genration varies much more than we suspected (Crippa et al. 2020 show this doesn't vary much by season in China), it would be very unlikely to get ratios larger/small than 0.8-1.2.

Thank you for making us aware of the study of Zhen et al. (2018). They showed that residential emissions have only a small contribution to Chinese emissions. However, this is an average value for the whole country and probably does not reflect the specific situation in cities. We checked the seasonality of the residential sector for China in Crippa et al. (2020) shown in Figure 1, which show only a small contribution of the residential sector but strong seasonal variability with more than two times higher $CO_2$ emissions in winter than in summer. This would of course only be an explanation for a strong seasonal emission ratio when it is a bigger part of emission source composition in Chinese cities than for China as a whole, which would be expected.

[Figure]

Figure 3 From Crippa et al., 2020 Seasonality of regional fossil $CO_2$ emissions in 2015 (expressed in Mt/month).

As described above, the updated wind and ozone data and the fact that our method reproduced the variability seen in the CEMS data have given us more confidence in our analysis and the results. Nevertheless, we now point out more strongly in the manuscript, that this is a widely used method, but the first time it is used also extensively on winter data and that there are more uncertainties involved. We have also added that our results to some part differ from previous literature, which however is mainly based on emission inventories, and that further detailed investigations and comparisons would be helpful to understand the discrepancies.

Line 381: I'm glad you are transparent with this, but these don't seem to be reasonable. NO2 lifetime should be shortest in summer, similar in Spring & Fall, and longest in Winter.

Yes, this is true, and in most cases, we do see this behavior, the shortest lifetimes in summer, the longest in winter, with spring and fall in-between and almost similar. Nevertheless, we also see, as in this case for Madrid, that there are deviations from the average and expected behavior. We see that lifetimes are not deviating a lot between seasons and already commented on possible reasons in the text (not yet sufficient statistics, clear-sky bias, midday

observation time of S5P, which all lead to more balanced lifetimes). We are now stating this more clearly in the text, especially for the case of Madrid in Figure 5a.

Line 391: Can't you go a step further and estimate OH for a few cities, perhaps from a model or previous literature, and see if the NO2 lifetimes you derive are approximately similar?

Unfortunately, we do not have OH model data available, but we checked previous literature. Lu et al. (2013) measured daily OH concentration maxima of 4-17e6 molec/cm$^3$ in Yufa, a suburban site south of Beijing, in summer 2006. With the assumption that the decay of NOx is determined by the termolecular reaction of OH with NO$_2$ (with the rate constant kOH+NO$_2$+M for 298K) this results in lifetimes of 1.5 h (17e6 molec/cm$^3$) to 6.3 h (4e6 molec/cm$^3$).

Dusanter et al. (2019) measured during the MCMA (Mexico City Metropolitan Area) field campaign in March 2006, maximum median OH concentrations of 4.6e6 molec/cm$^3$ during the day. The maximum OH concentration observed from day to day varied between 2e6 and 15e6 molec/cm$^3$. On a median basis OH peaked near 4.6e6 molec/cm$^3$ at noon, which would result in a lifetime of 5.5 h (1.7 h to 12.6 h).

This shows only two examples, one for summer measurements and one closer to winter in two different cities. Dusanter et al. (2019) compared their results to several other campaigns and concluded that OH measurements from winter are rare, that results for summer campaigns indicate that OH measurements are in a relatively small range of concentrations from 2e6 to 9e6 molec/cm$^3$, except for two campaigns where OH concentrations up to 20e6 molec/cm$^3$ were observed on some days. It shows how variable the OH concentrations can be between different measurement sites and even within one season.

The lifetimes estimated by OH concentrations from literature are similar to our lifetime estimates. It must be considered that the calculation of lifetime from OH concentrations is only a simplified assumption since NOx is not only lost through oxidation by OH in daytime (although this is the main process, especially during sunny conditions).

Line 415: Often, Saturday is different than Sunday… Saturday emissions can be quite larger than Sunday. See Goldberg et al., 2021 and Crippa et al., 2020 as an example

We added the following:
"Often one weekend day can be different than the other, for example, emissions on Saturdays are often not as low as those on Sundays, even if they are already lower than those from Monday to Friday (Crippa et al., 2020; Goldberg et al., 2021). If the weekend would be limited to one day in the analysis, this would lead to a strengthening of the weekend-to-week ratio for some cities."

Line 420: Also see Goldberg et al., 2021

Added Goldberg et al. (2021) as well as the explanation for deviations due to the different choice of weekend days.

Section 4.5 and Figure 7: If seasonal emissions are biased high in winter, then these drops attributed to COVID will be overestimated. Sections 4.2 & 4.3 need to revised in order for me to have more confidence in the results presented in this section.

In Section 4.5 we focus on comparing only identical months (for example April 2020 with April 2019). The different biases in different seasons should not affect this comparison much, since each month should have the same bias.

Section 4.6: Please more explicitly differentiate between random errors, which should mostly cancel out since you are averaging over many days of observations, and systematic errors, which would not cancel out.

We have changed the text to make it clearer which errors are systematic and which are random.

Line 564: Perhaps may even want to say that re-gridding TROPOMI data to a resolution of 0.1 degrees (~10 km) might be better than a higher resolution (~1 km) for this specific purpose only.

Yes, the Four Corners/San Juan power plant is a special case. We added an additional sentence:
"In this particular case, re-gridding the TROPOMI data to a coarser resolution would potentially allow an analysis"

References:
Crippa, M., Solazzo, E., Huang, G., Guizzardi, D., Koffi, E., Muntean, M., Schieberle, C., Friedrich, R. and Janssens-Maenhout, G.: High resolution temporal profiles in the Emissions Database for Global Atmospheric Research, Sci. Data, 7(1), 121, doi:10.1038/s41597-020-0462-2, 2020.
Goldberg, D. L., Lu, Z., Streets, D. G., De Foy, B., Griffin, D., Mclinden, C. A., Lamsal, L. N., Krotkov, N. A. and Eskes, H.: Enhanced Capabilities of TROPOMI NO2: Estimating NOx from North American Cities and Power Plants, Environ. Sci. Technol., 53(21), doi:10.1021/acs.est.9b04488, 2019.
Goldberg, D. L., Anenberg, S. C., Kerr, G. H., Mohegh, A., Lu, Z. and Streets, D. G.: TROPOMI NO2 in the United States: A detailed look at the annual averages, weekly cycles, effects of temperature, and correlation with surface NO2 concentrations, Earth's Future, e2020EF001665, doi:10.1029/2020EF001665, 2021.
Lu, Z., Streets, D. G., de Foy, B., Lamsal, L. N., Duncan, B. N. and Xing, J.: Emissions of nitrogen oxides from US urban areas: Estimation from Ozone Monitoring Instrument retrievals for 2005-2014, Atmos. Chem. Phys., 15(18), 10367–10383, doi:10.5194/acp-15-10367-2015, 2015.
Zheng, B., Tong, D., Li, M., Liu, F., Hong, C., Geng, G., Li, H., Li, X., Peng, L., Qi, J., Yan, L., Zhang, Y., Zhao, H., Zheng, Y., He, K. and Zhang, Q.: Trends in China's anthropogenic emissions since 2010 as the consequence of clean air actions, Atmos. Chem. Phys., 18(19), 14095–14111, doi:10.5194/acp-18-14095-2018, 2018

BEIRLE, Steffen, et al. Pinpointing nitrogen oxide emissions from space. *Science advances*, 2019, 5. Jg., Nr. 11, S. eaax9800.

DUSANTER, S., et al. Measurements of OH and HO 2 concentrations during the MCMA-2006 field campaign–Part 1: Deployment of the Indiana University laser-induced fluorescence instrument. *Atmospheric Chemistry and Physics*, 2009, 9. Jg., Nr. 5, S. 1665-1685.

LU, K. D., et al. Missing OH source in a suburban environment near Beijing: observed and modelled OH and HO 2 concentrations in summer 2006. *Atmospheric Chemistry and Physics*, 2013, 13. Jg., Nr. 2, S. 1057-1080.

---

## Author Comment (AC2)

Comments from anonymous Referee #2:

We would like to thank the reviewer for his/her helpful comments. We hope that we could address all questions and unclear points satisfactorily.

In the course of the revision, we have made the following important changes to the analysis: We updated our wind and ozone data and are now using ERA5 reanalysis data averaged about the boundary layer instead of a fixed height to include possible seasonality in these input data. This led to changes in all our estimates but not in any fundamental way. We also applied our method to a power plant for which hourly NOx emissions are reported, and analyzed the temporal variability over the course of the year in more detail. From this comparison we conclude that the EMG method applied to TROPOMI data at least for this power plant can reproduce the temporal variability reasonably well and does not show a clear seasonal bias. To get a better representation of source areas from Southern Hemisphere, we included five new target areas. The results fit well into the overall result of the study.

Legend: Referee comments in black, author comments in blue

**Review "Variability of nitrogen oxide emission fluxes and lifetimes estimated from Sentinel-5P TROPOMI observations", by Lange et al.**

The study by Lange et al. applies the EMG method to TROPOMI $NO_2$ observations to estimate NOx emissions and lifetimes in various urban areas and sources. They study in detail the variability, seasonality, trends and geographical dependence of the NOx emissions and lifetimes. The method is robust and has been applied to satellite data in the past, with the added value of detailed analysis and other sources being analyzed in this study. The paper is well written and well structured, with quality figures supporting the methods and discussion provided in the text.

Although results and conclusions are presented in a clear way, there are some points that need to be addressed to make the conclusions more substantial and robust. So, I would recommend for publication once the comments below are addressed by the authors.

**General comments**

In general, the introduction could be improved. Currently the concepts are mixed, and it is not to the point. After reading it, it is not clear what the study is going to be about, and what the research adds to the current literature that is discussed is lacking. It starts with some chemistry, then lifetime, emission estimates, seasonal, day to day, trends, then covid and then back to lifetimes to then introduce the satellite measurements that have already been mentioned at different points in the text before. In the last paragraph you explain what you do in the study but still go back to literature, so not to the point. I think the text and ideas are there, but somewhat disorganized.

Thank you for this feedback. We went through the introduction and structured it more clearly in the different parts and better highlighted what we do in this study. We hope it is now more to the point.

Sect. 4.5 and the discussion on the Covid effects on NOx emissions is vague. You argue that EMG method accounts for meteorology and other possible effects on NOx emissions, but the arguments do not hold equally for the three cities that are discussed and different months. Only months where results are supported by the covid hypothesis are brought forward, which makes conclusions less solid. Many cities have been analyzed in literature to study covid effects on air pollution, so the fact that only three are presented here is also a sign that the method does not work in other cities. If this is not the case, then this should also be stated in the manuscript.

Yes, this is right. We cannot use our method for many cities. We also discussed this in the manuscript. Since the EMG method can only be used to investigate isolated point sources, the range of possible study areas is limited. In addition, we are also much more limited compared to the rest of the study, because we decided to compare only monthly means from the two different years with each other to minimize influence by meteorology. Even if the method takes wind conditions into account, other things like temperature still play a role. In addition, Covid regulations changed quite quickly, and changes can be better detected when shorter time periods are compared.

However, monthly averages bring the disadvantage that sometimes only a few days are available for averaging and analysis due to cloud cover. If only a few days are available in a month, the statistics are simply missing and, for example, weekday effects also have significant influence. This is also the reason why we cannot consider all months even for the cities selected here, and it has to be considered that the statistics may not be ideal. In this context, we would like to point out that some earlier studies use data with less strict quality filtering, including also measurements under cloudy conditions which have intrinsically much larger uncertainties.

We are also discussing months which are not fitting in the expected behavior, as for example for Buenos Aires the June with clearly higher emissions in 2019 than 2020, in this case temperature can be a possible explanation (see also comment below).

The results discussed in other sections may be affected by Covid, but there is no mention to that. How are the summer to winter ratios affected by covid? How is the comparison to EDGAR affected by the supposedly lower emissions when restrictions due to covid where active?

All results with exception of the COVID chapter are based on data from March 2018 to February 2020, basically excluding data for which COVID regulations changed emissions. However, we agree that we should be more careful with our results for Chinese cities since restrictions due to COVID started in Wuhan on 23 January 2020 followed by more Chinese cities and then next northern Italy on 8 March. Therefore, of the regions we analyzed, the cities of Wuhan and Xian in China are potentially already affected by COVID regulations when analyzing data from March 2018 to February 2020. Therefore, we repeated the analysis for these two cities and limited the data period from March 2018 to 22 January 2020.

For Wuhan this resulted in a change of estimated winter emissions from $151.1 \pm 20$ mol/s to $254.9 \pm 39$ mol/s but of course also to reduced statistic due to a reduction of available days from 16 to 9 days in average over the target area during winter. With this, the summer-to-winter ratio changed from 0.5 to 0.3.

For Xian we could not perform a seasonal and weekday analysis of emission due to insufficient data availability caused by filtering the data for clouds and quality.

Sect. 4.6: The 43-63% uncertainties are higher than the 1-sigma uncertainty provided by the EMG method (e.g. Table 1 uncertainties are ~ 5 %). The high uncertainties need to be reflected in the uncertainty values you provide for the emissions and lifetimes, and discussed in this section.

We commented in Section 3.3 on the fact that our given uncertainties and error bars for emission and lifetime estimates are only based on 1-sigma uncertainties provided by the EMG method and that uncertainties in general are discussed in more detail in section 4.6. We have added the following, to clarify that the uncertainties discussed in section 4.6 are much higher than the 1-sigma from the fit:
"Since the estimates are influenced by additional error sources, dominated by the accuracy of the TROPOMI $NO_2$ tropospheric vertical column and the wind field, uncertainties in general are significantly higher, ranging from 43-63%. More details can be found in section 4.6."
In addition, we added some small comments to the manuscript when critical results are discussed.

**Specific comments**

**Introduction**
Line 101: Is not only urban areas that are the target of the study, right? Same at the beginning of Sect. 5.

Yes, indeed not only urban areas were investigated. For the first case we just deleted the "urban" in the text, for the second case we changed it to "NOx source areas".

Why you do not discuss Beirle et al. (2021) in the Introduction? Is it because those are only point sources?

Thank you for pointing out that Beirle et al. (2021) is missing, we included it now in the newly structured introduction.

**Data**
Line 154 when referring to Fig. 1: "The red circles mark regions with higher $NO_2$ than their surroundings and are analyzed in this study." Is this the reason why you choose these regions? Now it reads as it would be like that, but there are many more places with "higher $NO_2$ than the surroundings".

We agree that this was misleading. We skipped the reference in this section and added the figure to section 3.1., where also the guideline for the selection of targets is described, and changed the text to "we selected 45 target regions, which are marked with red circles".

Wind data and ozone mixing ratio: one is interpolated to the exact overpass time and the other to "typical mean early afternoon overpass time". Why this difference?

As part of the update to the new wind data, we also updated the ozone data which we now also derive from the ERA5 reanalysis data which have a better spatial resolution and much better temporal resolution. Ozone data are averaged over the boundary layer just as wind data and thereby provide a better representation of seasonal variability. We use now hourly

ozone volume mixing ratios with a horizontal resolution of 0.25° in model levels averaged over the boundary layer and interpolated to TROPOMI overpass time and oversampled to the same 0.01° resolution as the TROPOMI data.

Line 191: were "new sources" found in Georgoulias et al. (2019)? Or was it positive trends as well as negative ones, that also reversed during the last decades? Could you use the trends from 2015-2017 from this study to extrapolate EDGAR inventory?

You are right, this formulation was a bit misleading. The text now is:
"2015 is the most recent year available in EDGAR v5.0, which cannot reflect either the recent negative as well as positive trends, which were found in trend analysis of NO$_2$ column satellite data (Georgoulias et al., 2019). Since a large part of the regions analyzed in this study is located in industrialized and highly populated regions, where NO$_2$ has generally decreased according to Georgoulias et al. (2019), we anticipate that the TROPOMI estimates for the majority of the analyzed regions are lower than the EDGAR inventory estimates for 2015"

Georgoulias et al. found positive as well as negative trends during the last 2 decades (April 1996 – September 2021). Tropospheric NO$_2$ has generally decreased during the last 2 decades over the industrialized and highly populated regions of the heavily industrialized world and increased over industrially more developing. It is suggested in the reference that linear trends cannot be used efficiently worldwide. Tropospheric NO$_2$ is very sensitive to socioeconomic changes which may cause either short-term variations or even a reversal of the trends. Different and recent trends would be needed for each city, which is not available.

Sect. 3.1.: why you do not choose any region in Australia? Southern Hemisphere is underrepresented in your selection.

We checked for more possible source areas from Southern Hemisphere and added the following to our study:

Adelaide (Australia)
Brisbane (Australia)
Hwange (Zimbabwe)
Melbourne (Australia)
Perth (Australia)

Pleases mention here the > 2m/s applied to the overall filtering criteria.

Done: "Regions having few clouds are preferred to maximize the number of satellite observations. In addition, a local meteorology for the target region, having rather homogeneous wind patterns, facilities the detection of the outflow patterns, which is further enhanced by filtering out data with wind speeds of less than 2m/s (Beirle et al., 2011)."

Have you looked at boundary layer height information? 100 m is probably too low, so you might introduce a systematic error here.

We changed our wind data to wind speed and direction averaged in the boundary layer, extracted from ERA5 re-analysis model level data. Therefore, we adapted the section about

the wind data and included that the chosen height of wind data can also be critical for wind direction. To exclude the possibility that the strong seasonality in the emissions is due to a lack of seasonality in the wind data (seasonal variability of boundary layer height -> higher/lower wind speed), we decided to use an average over the boundary layer instead of the 100 m data.

**Results and discussion**
Sect. 4.1 & Sect. 4.2:

How do your summer estimates agree with Goldberg et al. (2019)?

We added results for the same time period (April to September 2018) in brackets to the table and some additional comments to the text. Using the same time periods gives a better agreement with Goldberg et al. (2019), deviations are much smaller and could be caused by different wind and ozone data which have quite a big influence. An additional difference is that data are manually rotated in Goldberg et al. (2019) while we use ERA5 wind data for this purpose.

Are there any temperature anomalies in the areas you study that could point to an over/under estimation of the emission inventories based on the threshold temperatures assumed for heating/air conditioning?

Unfortunately, we do not have enough information about temperature anomalies in the different areas and for the relevant years (2015,2018-2020), as well as how exactly heating and cooling degree days are considered for the individual countries in EDGAR, to analyze this.

Sect. 4.3: Lifetimes need to be validated in order to discard unrealistic effects on the NOx emissions and ratios shown in Sect. 4.2. Also, in view of the last paragraph of Sect .4.4. Or at least make clear that the uncertainties associated to the lifetimes is much higher than 1-sigma given by the method, which results in important uncertainties in the emission estimates.

We already compare our lifetimes to existing studies in the manuscript. In response to the reviewer's comment, we now discuss this more and also put more emphasis on the uncertainties in the lifetime estimates and classify what that means for our results.

Sect.4.4: line 420: what if you only take into account summer data as in Goldberg et al. 2019?

We found out that weekend in Goldberg et al. was defined only as Sunday and we defined weekend as Saturday and Sunday which explains the big discrepancies between Goldberg et al. (2019) and our results as Goldberg et al. (2021) and Crippa et al. (2020) showed that emissions on Saturdays are often not as low as those on Sundays, even if they are already lower than from Monday to Friday.

Sect. 4.5: line 469: this can be known by looking at meteorology anomalies in this period.

We checked ERA5 re-analysis temperature data averaged over the boundary layer in the Buenos Aires target area for the measurement days used in our study of the two years 2019 and 2020 for the months May, June and July. The month of June 2020 was 3°C colder than the

year before. For May and July 2019 and 2020 the temperatures are very comparable. The colder temperatures in June 2020 than in June 2019 are still not a clear causal factor for the unexpectedly higher emissions in June 2020 compared to June 2019 despite the COVID pandemic but support this possible explanation. We added a comment to the text.

| | Mean temperature (K) | |
|---|---|---|
| | 2019 | 2020 |
| May | 288.4 K | 288.3 K |
| **June** | **285.7 K** | **282.8 K** |
| July | 281.8 K | 281.2 K |

Line 475: two years cannot be considered a "trend". Has the economy changed so much that is noticeable in emissions?

Agreed. We have now formulated this more carefully:

"In January and February, the calculated NOx emissions are higher in 2020 than in 2019. There is no impact of Covid-19 yet in this period, and India's fast-growing economy is probably the explanation for the upward trend in NOx emissions."
Is changed to:
"There is no impact of Covid-19 yet in this period, and the higher emissions in 2020 compared to the year before match the fact that India has a fast-growing economy. Georgoulias et al. (2019) showed a positive trend of 3.1±0.5 % per year for the period April 1996 to September 2017."

Line 483: This would indicate that the lockdown was much stricter in May compared to October, so these estimations might be affected by other factors rather than covid, which could point to the lifetime that is estimated being too low.

We assume you meant it the other way round, because we found higher reductions in October than in May.
The strictest lockdown for Madrid was from 28 March to 12 April, then some restrictions were lifted from 13 April to 1 May, and in May the government followed a plan for easing lockdown restrictions back to normal. On 1 October the government ordered a partial lockdown for Madrid again. It therefore might be the case that the lockdown was stricter in October than in May but of course, this is difficult to quantify.
It is difficult to exclude that other factors besides Covid have had an influence. The influence of seasonal lifetime issues should be small as we look at relative changes between the same months only (October 2020 to October 2019 and May 2020 to May 2019, respectively). Statements about absolute values should be viewed with more caution, which is why we do not make any.

What is the explanation for higher emissions in Feb. 2020 compared to 2019?

Do you still refer to Madrid, there we see it the other way round than in your comment, but we assume you meant it, because this strong deviation between the two months without the influence of regulations due to COVID are striking as they are not expected. We think that the

significantly higher emissions in February 2019 compared to 2020 are mainly caused by two factors, first a strong synoptic meteorological variability in Europe and second February is also a month with typically persistent cloud cover, resulting in a reduced number of TROPOMI observations which will reveal more natural variability. Similar findings are also described in Bauwens et al. (2020) and Levelt et al. (2021).

**Technical corrections**
Throughout the manuscript, please modify "nitrogen oxide" by "nitrogen oxides" and "emission" by "emissions" when necessary.

We checked our manuscript for this and changed it accordingly.

NOx is plural, so NOx are emitted (e.g. page 1, line 15).

We checked our manuscript for this and changed it accordingly.

Please define acronyms first time they are mentioned (e.g. abstract lines 1-2 TROPOMI, NOx)

Done.

Abstract: "is ECMWF ERA 5" relevant? Just "wind data" could be sufficient, as there is no need to go into details in the abstract.

We deleted ECMWF ERA5 from the abstract.

Line 90: "but analyses with GEOS-Chem…": reader does not necessarily know what GEOSChem is, so please specify.

Thank you for the comment, we have added in the text that it is a chemical transport model.

Line 92: do you mean "measurements of $NO_2$" instead of "measurements of NOx"?

Changed NOx to $NO_2$.

Line 116: "(" missing

Corrected.

Line 152: "removes part of the scenes" -> "removes the scenes"

Thank you for pointing out, this is misleading, the sentence was corrected to:
"A qa_value of 0.75 removes problematic retrievals, errors, partially snow/ice covered scenes and measurements with cloud radiance fractions of more than 50%, which roughly corresponds to a geometric cloud fraction of 0.2."

Line 215: facilities - > facilitates

Done.

References:

BAUWENS, M., et al. Impact of coronavirus outbreak on NO2 pollution assessed using TROPOMI and OMI observations. *Geophysical Research Letters*, 2020, 47. Jg., Nr. 11, S. e2020GL087978.

GEORGOULIAS, Aristeidis K., et al. Trends and trend reversal detection in 2 decades of tropospheric NO 2 satellite observations. *Atmospheric Chemistry and Physics*, 2019, 19. Jg., Nr. 9, S. 6269-6294.

GOLDBERG, Daniel L., et al. Enhanced Capabilities of TROPOMI NO2: Estimating NO X from North American Cities and Power Plants. *Environmental science & technology*, 2019, 53. Jg., Nr. 21, S. 12594-12601.

LEVELT, Pieternel F., et al. Air quality impacts of COVID-19 lockdown measures detected from space using high spatial resolution observations of multiple trace gases from Sentinel-5P/TROPOMI. *Atmospheric Chemistry and Physics Discussions*, 2021, S. 1-53.

---

## Author Comment (AC3)

Comments from anonymous Referee #3:

We would like to thank the reviewer for his/her helpful comments. We hope that we could address all questions and unclear points satisfactorily.

In the course of the revision, we have made the following important changes to the analysis: We updated our wind and ozone data and are now using ERA5 reanalysis data averaged about the boundary layer instead of a fixed height to include possible seasonality in these input data. This led to changes in all our estimates but not in any fundamental way. We also applied our method to a power plant for which hourly $NO_x$ emissions are reported, and analyzed the temporal variability over the course of the year in more detail. From this comparison we conclude that the EMG method applied to TROPOMI data at least for this power plant can reproduce the temporal variability reasonably well and does not show a clear seasonal bias. To get a better representation of source areas from Southern Hemisphere, we included five new target areas. The results fit well into the overall result of the study.

Legend: Referee comments in black, author comments in blue

The study "Variability of nitrogen oxide emission fluxes and lifetimes estimated from Sentinel-5P TROPOMI observations" by Lange et al. presents emission and lifetime estimates from TROPOMI NO2 measurements for a selection of source regions around the world, and investigates weekly & seasonal cycles as well as the impact of recent lockdown measures.

The manuscript is clearly written, provides a detailed description of method & error calculation, and discusses the results adequately, also in context with previous studies.
I recommend publication after dealing with the comments below.

NOx: x should be subscript throughout the manuscript.

NOx was changed to $NO_x$ throughout the manuscript.

Line 3: this resolution was not provided in the first months of operation.

We changed the sentence to: "In this study, two years of TROPOMI $NO_2$ data, having a spatial resolution of up to 3.5km x 5.5km, have been analyzed together with wind data." More details about the change of resolution are given in section 2.1.

Line 7: "dessert" should be "desert"

Changed.

Lines 14&15: 2x "In the atmosphere". I would skip the first one.

We skipped the first one.

Line 123: Please specify that for the highest latitudes considered here, up to two overpasses are available. Three overpasses only occur for higher latitudes.

For some days there are three overpasses available, not every day and two of them are then also only on the outer edge of the orbit but for example on 17 July 2020 there are three overpasses available for Saint Petersburg:

| Date | UTC | Satellite azimuth | Satellite elevation | Viewing zenith | Solar zenith | Satellite direction | Distance to site (km) |
|---|---|---|---|---|---|---|---|
| 17.07.2020 | 08:41 | 47.23 | 25.65 | 64.35 | 41.41 | ascend | 1287.63 |
| 17.07.2020 | 10:20 | 69.08 | 83.41 | 6.59 | 38.82 | ascend | 85.91 |
| 17.07.2020 | 12:02 | 270.67 | 25.23 | 64.77 | 43.94 | ascend | 1305.12 |

We rewrote the sentence to make clear that two of the three measurements are then on the edge of the orbit: "At higher latitudes, on some days up to three overpasses, with approximately 100 minutes in between the measurements, are available. In the rare case of three overpasses, two of them are only with the outer edge of the swath."

Lines 126-129 (up to "Seo et al."): This is off-topic and should be skipped.

We skipped the part about the other available products of TROPOMI.

Line 135: Please explain why the difference in resolution is more extreme at the edges.

The larger pixels are a geometric effect due to the slanted view to the ground. To avoid this, for TROPOMI, in the middle part of the scan always two pixels are averaged but not at the edges. As a result, the resolution decreases towards the edges, improves by a factor of two and then decreases again. We added a short comment to the manuscript.

Lines 145ff: Please explain why versions up to 1.03 are mixed in your analysis, but 1.04 is excluded.   Shortly mention what has changed in 1.04 and refer to the discussion of the low bias of tropospheric NO2 in section 4.6.

We made some changes in the text to better explain the version changes:

"Changes between the versions before 1.04. are only minor and data can be mixed. 1.04 was activated on 02 December 2020 and implemented major changes, which led to a substantial increase in the tropospheric $NO_2$ column over polluted areas for scenes with small cloud fractions. We only use the data up to end of November 2020 for our analysis to ensure better comparability, since mixing data from before version 1.04. and after is not recommended. A complete mission reprocessing will be performed to get a harmonized dataset (Eskes and Eichmann, 2021). "

We also added some more discussion in section 4.6 about the version changes and possible influence on the analysis when using the reprocessed dataset.

Line 153: Do you have a reference for 50% CRF ~ 20% GCF?

We do not have a reference and deleted the last part about the GCF, as it is not important for the study.

Line 154: I am confused by the different time intervals.
In line 144: March 2018 to November 2020.
In line 197: March 2018 to February 2020 general January 2019 to November 2020 Covid 19
Why is yet another interval chosen for Fig. 1? Why not March 2018 to February 2020?

We changed the time interval for Fig1 to the complete time interval March 2018 to February 2020 of the main analysis.

For the main part we used data from March 2018 to February 2020 to exclude the influence of COVID-19 regulations. Following a comment from referee 2, we now shortened the time interval for the two analyzed cities in China (Wuhan, Xian) since lockdown in China started on 23 January 2020. All other cities should not be influenced since all following lockdowns were after February, the next one was in northern Italy on 8 March.

For the Covid analysis we used January 2019 to November 2020 data.

Line 155: The criterion "higher NO2 than their surrounding" would result in a quite different selection of hot spots visible in the mean map. I propose to skip the reference to Fig. 1 in this section, but add it section 3.1, where the selection of source regions is explained in detail.

Thank you for the comment, we skipped the reference in this section and added the figure to section 3.1.

Caption Fig. 1: Add a reference to table A1 for further details.

Done

Line 162: Please quantify "low".

We added more detail here: "The resulting emissions and lifetimes change less than 2% (5%) on average when using 200m (1000m) instead of 500m and less than 15% for individual sources."

Section 2.3: Using model input in order to correct for the NO2/NOx ratio is probably an improvement compared to just taking a constant value. However, I don't understand why the calculation in Eq. 3 was done for a fixed pressure, since the model would also provide the actual pressure at the altitude of interest (probably the same as chosen for the wind vector). The considered source regions include some elevated locations like Las Vegas, Medupi, or Colstrip (1 km asl!), where 950 hPa is not appropriate.

We updated the ozone data and are now also using the ERA5 reanalysis data. We use hourly ozone volume mixing ratios with a horizontal resolution of 0.25° in model levels averaged over the boundary layer and interpolated to TROPOMI overpass time and oversampled to the same 0.01° resolution as the TROPOMI data.

Line 219: How far does the - somehow arbitrary - selection by visual inspection affects the generality of the results? Could there be some "selection bias"?

This is of course hard to quantify. Of course, we tried to select a good mix of sources from different regions, we extended our analysis region as far as possible north. Referee 2 suggested to add more source regions from the southern hemisphere, and we added 5 new target regions. We analyzed urban sources, isolated power plants, oil fields, industrial regions and some mixed sources. Our results show a wide range of mean emissions $(4.6 \text{mol/s} - 272.9 \text{mol/s})$ and lifetimes $(1.2h - 8.2h)$. Nevertheless, some selection bias could have been introduced, for example by not analyzing regions with low wind speed or areas in regions with many emission sources and

higher background levels as for example in eastern China. We added a comment about this possibility in this section and also in the conclusion.

Line 224: What amount of TROPOMI observations is needed in order to get robust results from this method?

The method works in principle on a single TROPOMI overpass as shown in Lorente et al. (2019) and Goldberg et al. (2019). If the conditions are favorable (e.g clear sky, homogeneous, strong wind conditions…) a single overpass can deliver good results.
But of course, it depends on the goal of the analysis. We wanted to analyze variability due to seasonal or weekday effect in a more general way, therefore we need better statistics to not for example introduce a bias in the seasonal analysis by weekday variability. We used two years of data for our analysis, to have for example at least two times three months for seasonal analysis. This also makes it possible to analyze regions which are covered by clouds more frequently. We added a comment to the text, that shorter time intervals are possible depending on the goal of analysis.

Line 293: What is "dispersion" meaning in this context?

"effective mean dispersion lifetime" – in this case we mean dispersion (e.g. dilution) by wind.

Line 294: or the effects of non-linearities in the NOx lifetime.

Added to the text.

Figure 5: This figure is very hard to read. I propose to add one further panel, highlighting the seasonal dependence. E.g. show the lifetime for different locations as function of season.
     - arrange the 4 panels 2x2
     - enlarge

We created an additional figure, which shows the lifetime for different locations as function of season and arranged all 4 plots in 2x2. The additional figure is showing the seasonal dependence in more detail and the new arrangement makes everything better to read. In order to keep the additional figure easy to read, we could only include a selection of the sources we analyzed. We tried to find a mix of regions from southern and northern latitude, cities as well as powerplants and well distributed over many latitudes

[Figure]

Line 430: One important difference is that the tropospheric columns also contain the upper tropospheric background, which is removed in the EMG method by fitting the background B.

Thank you for pointing this out, we added this comment to the text.

Lines 507ff: This section is quite vague. One main reason for the low bias was identified as the FRESCO cloud height which is biased low (Compernolle et al., https://amt.copernicus.org/articles/14/2451/2021/) causing high-biased AMFs. This was changed with v1.04 of the operational NO2 product. Please extend this discussion accordingly.

We added more details and discussion about the version changes and possible influence on the analysis when using the reprocessed dataset in section 4.6.

Line 613: Please specify "short time periods": how many TROPOMI overpasses are needed?

We added the following comment about possible time periods:
"Depending on the goal of the analysis, already a few days of measurements can be sufficient; for seasonal studies, depending on the local meteorological conditions, one to two seasons are sufficient, and it is not anymore necessary to average over several years."